# mTOR activity is essential for retinal pigment epithelium regeneration in zebrafish

Fangfang Lu[1,2], Lyndsay L. Leach[1], Jeffrey M. Gross[1,3,4]*

**1** Department of Ophthalmology, Louis J. Fox Center for Vision Restoration, University of Pittsburgh School of Medicine, Pittsburgh, Pennsylvania, United States of America, **2** Department of Ophthalmology, The Second Xiangya Hospital, Central South University, Changsha, Hunan, China, **3** Department of Developmental Biology, University of Pittsburgh School of Medicine, Pittsburgh, Pennsylvania, United States of America, **4** Department of Molecular Biosciences, The University of Texas at Austin, Austin, Texas, United States of America

\* grossjm@pitt.edu

**Data Availability Statement:** Raw read and processed data files are available in GEO under accession number GSE174538. https://www.ncbi.nlm.nih.gov/geo/query/acc.cgi?acc=GSE174538.

## Abstract

The retinal pigment epithelium (RPE) plays numerous critical roles in maintaining vision and this is underscored by the prevalence of degenerative blinding diseases like age-related macular degeneration (AMD), in which visual impairment is caused by progressive loss of RPE cells. In contrast to mammals, zebrafish possess the ability to intrinsically regenerate a functional RPE layer after severe injury. The molecular underpinnings of this regenerative process remain largely unknown yet hold tremendous potential for developing treatment strategies to stimulate endogenous regeneration in the human eye. In this study, we demonstrate that the mTOR pathway is activated in RPE cells post-genetic ablation. Pharmacological and genetic inhibition of mTOR activity impaired RPE regeneration, while mTOR activation enhanced RPE recovery post-injury, demonstrating that mTOR activity is essential for RPE regeneration in zebrafish. RNA-seq of RPE isolated from mTOR-inhibited larvae identified a number of genes and pathways dependent on mTOR activity at early and late stages of regeneration; amongst these were components of the immune system, which is emerging as a key regulator of regenerative responses across various tissue and model systems. Our results identify crosstalk between macrophages/microglia and the RPE, wherein mTOR activity is required for recruitment of macrophages/microglia to the RPE injury site. Macrophages/microglia then reinforce mTOR activity in regenerating RPE cells. Interestingly, the function of macrophages/microglia in maintaining mTOR activity in the RPE appeared to be inflammation-independent. Taken together, these data identify mTOR activity as a key regulator of RPE regeneration and link the mTOR pathway to immune responses in facilitating RPE regeneration.

## Author summary

Age-related macular degeneration (AMD) is a leading cause of blindness world-wide, with incidences predicted to rise substantially over the next few decades. Cells of the retinal pigment epithelium (RPE) are affected in AMD and there are currently no effective

**Funding:** The work was supported by the National Institutes of Health NIH R01 EY29410 and NIH CORE Grant P30 EY08098 to JMG; the University of Pittsburgh Medical Center Immune Transplant and Therapy Center to JMG and LLL. This work was also supported by an unrestricted grant from Research to Prevent Blindness. FL was supported by a research fund from the Xiangya Medical School, Central South University and China Scholar Council for studying in Pittsburgh. The funders had no role in study design, data collection and analysis, decision to publish or preparation of the manuscript.

**Competing interests:** The authors have declared that no competing interests exist.

therapies that slow RPE cell death or restore lost RPE cells in advanced-stage AMD. An exciting potential approach to treat many diseases of the eye, including AMD, is to stimulate endogenous regeneration to restore cells lost to disease. For this to become possible, we must first understand the molecular and cellular underpinnings of the regenerative response. In this study, we utilize zebrafish as a model system, which possess tremendous regenerative potential in multiple cell and tissue types, including the RPE. Our results identify the mTOR signaling pathway as a key regulator of RPE regeneration. We identify a link between mTOR signaling and immune responses, which are known to modulate regeneration of a variety of tissues and organs. Together, our results identify one of the first molecular mechanisms facilitating intrinsic RPE regeneration and these data could serve as a foundation for the development of new therapies aimed at stimulating the regeneration of RPE cells in the diseased eye.

## Introduction

The retinal pigment epithelium (RPE) is a monolayer of ocular cells located between the vascular choriocapillaris and neural retina. The RPE plays myriad critical roles during vision and this is highlighted by the prevalence of degenerative blinding diseases like age-related macular degeneration (AMD), in which the RPE is primarily affected. AMD is a leading cause of blindness in the developed world; indeed, advanced AMD with geographic atrophy (so called "dry" AMD) accounts for the majority of AMD cases worldwide [1]. Functions of the RPE during vision include: phagocytosis of photoreceptor outer segments and recycling of key components of the visual cascade; transport of nutrients and ions between the choriocapillaris and retina; secretion of growth factors; and contributing to the blood-retina barrier, amongst others [2]. Despite the importance and involvement of the RPE in human blinding diseases, there are currently no FDA-approved treatments available to treat RPE loss in diseases like AMD.

In recent years, RPE cell replacement therapies like transplantation of embryonic or induced pluripotent stem cell-derived RPE, autologous peripheral RPE, or RPE progenitor cells have been shown to restore some visual function in animal models of human disease, and some are in early-phase human trials [3]. For example, clinical trials with RPE patches derived from several different cell sources have revealed preliminary success in safety and efficacy, long-time survival, and improvement in best-corrected visual acuity in AMD patients [4–8]. Despite this, these therapies have not yet been proven effective enough for use as an AMD treatment, and immune rejection, toxicology of RPE transplants and function of the grafts *in vivo* are challenges that remain to be overcome before these can be clinically available [9]. A complementary reparative strategy to explore for use in human patients is to stimulate intrinsic RPE regeneration. However, in mammals, the RPE shows an extremely limited capacity to regenerate after injury [10–13]. RPE cells terminally differentiate during the early stages of eye development and remain mitotically inactive in adults, under most conditions. RPE cells do have a modest capacity to proliferate in some injury contexts; however, this can turn pathological in conditions like proliferative vitreoretinopathy [14]. Recently, it was shown that a subpopulation of adult human RPE cells, termed retinal pigment epithelial stem cells (RPESCs), were capable of self-renewal and differentiation into neural and mesenchymal progeny *in vitro* [15,16]. This was an exciting result as it demonstrated that there could be an intrinsic, but latent, potential for human RPE to be induced to regenerate.

In contrast to mammals, zebrafish possess remarkable regenerative capacity after injury in multiple tissues [13,17], including the RPE [18,19]. Previous work from our laboratory

established a transgenic zebrafish RPE ablation model (*rpe65a*:nfsB-eGFP) and demonstrated that RPE could intrinsically regenerate into a functional monolayer after widespread genetic ablation [18]. Using this model, Wnt signaling and innate immune system activity have been shown to facilitate RPE regeneration [18,19]. Despite these advances, the molecular mechanisms underlying intrinsic RPE regeneration in zebrafish are still unclear. Identifying these mechanisms could have a significant impact on developing strategies to stimulate endogenous regeneration in the human eye.

The mechanistic target of rapamycin (mTOR) pathway is activated by a number of intracellular and extracellular inputs that include growth factors, stress, energy, oxygen, and nutrient levels to regulate cell growth and metabolism during development and in a variety of physiological processes [20,21]. mTOR activity is also known to contribute to RPE function. For example, mTOR is activated in the RPE physiologically by the phagocytosis and digestion of photoreceptor outer segments [22–24] and pathologically by stress, which leads to RPE cell dedifferentiation and hypertrophy [25]. Moreover, *in vitro* studies using cultured RPE have demonstrated that mTOR activity is essential for the survival, migration and proliferation of RPE cells after damage [26–28]. mTOR activity is also associated with regeneration in several organs and tissues across Metazoa [29]. For instance, in zebrafish, mTOR activity regulates fin [30], liver [31], and heart regeneration [32]. In mammals, mTOR plays important roles in optic nerve regeneration [33–35] and this function is conserved in human induced pluripotent stem cell (iPSC)-derived retinal ganglion cells [36]. In zebrafish and chicken, mTOR activity has also been shown to facilitate retinal regeneration [37,38].

Functions of mTOR during RPE regeneration have not been reported but given the association with regenerative events in other contexts, and known functions in the RPE, we hypothesized that mTOR might facilitate RPE regeneration. Taking advantage of our zebrafish RPE ablation-regeneration model, here we demonstrate that mTOR is rapidly activated in RPE cells post-ablation. Through pharmacological and genetic manipulations, we show that mTOR activity is necessary for RPE regeneration and that ectopic activation of the mTOR pathway expedites RPE regeneration post-injury. Mechanistically, mTOR activity after RPE injury triggers the recruitment of macrophages/microglia to the injured RPE, and these leukocytes then further reinforce mTOR activity in the regenerating RPE. Taken together, these data highlight a novel role for mTOR during RPE regeneration and reveal another avenue through which therapies could be developed to restore RPE lost to degenerative ocular disease.

## Results

### Dynamics of mTOR activity after RPE ablation

To stimulate RPE injury and subsequent regenerative responses, we utilized *rpe65a*:nfsB-eGFP transgenic zebrafish and metronidazole (MTZ)-mediated cell ablation[18]. In this model, MTZ addition results in ablation of approximately the central two-thirds of the RPE, which stimulates a regenerative response over the following two weeks. After onset of RPE ablation at 5 days post-fertilization (dpf), the spatial and temporal activity of mTOR was quantified using a monoclonal antibody against phosphorylated 40S ribosomal protein S6 (p-S6) at residue Ser235/236. S6 is a downstream effector of the mTOR pathway and phosphorylation at Ser235/236 is routinely used as a readout of mTORC1 activation [31,39]. We focused on post-injury time points ranging from 3, 6, and 12 hours post-injury (hpi) to 4 days post-injury (dpi), which correspond to 5.125, 5.25, and 5.5 days post-fertilization (dpf; referred to as MTZ⁻ uninjured (UI) for simplicity) to 9dpf (Fig 1A), which encompass the peak phases of the RPE regenerative response [18]. In MTZ⁻ larvae, low basal levels of p-S6 were detected in the RPE layer from 3h to 12h in UI time points, and p-S6 was almost undetectable in the RPE layer

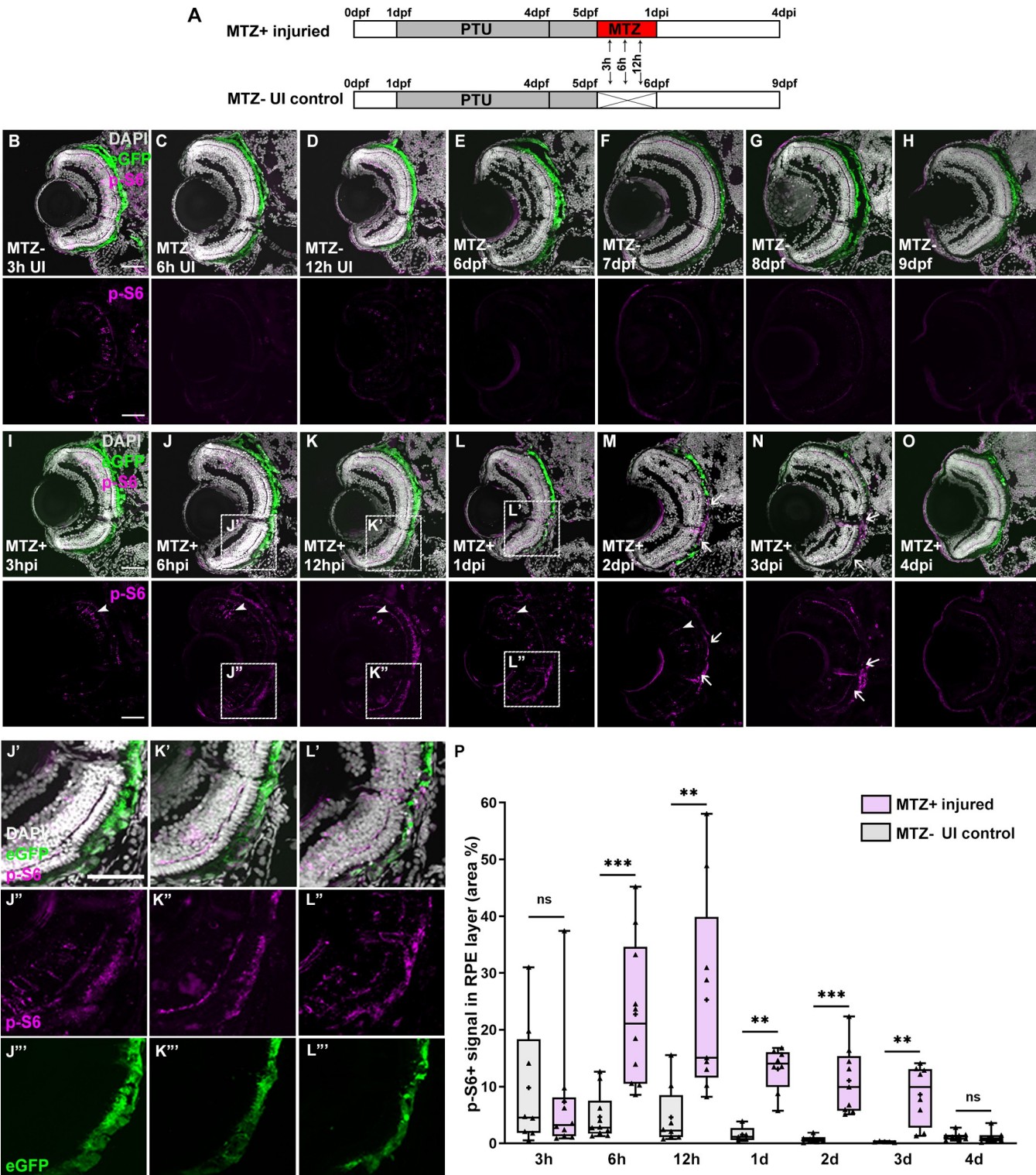

**Fig 1. The mTOR pathway is activated in the RPE after ablation.** (A) Schematic of the experimental paradigm showing the time points for MTZ ablation and sample collection on transgenic larvae (*rpe65a*:nfsB-eGFP). (B-H) Immunofluorescent images of p-S6 staining on transverse cryosections from unablated (MTZ⁻) and (I-O) ablated (MTZ⁺) larvae from 3hpi to 4dpi. *White arrows* indicate the p-S6⁺ eGFP⁻ cells in the RPE layer. *White arrowheads* indicate p-S6⁺ cells in the retinae. (J'-J''') High-magnification images showing the colocalization of p-S6 and eGFP at 6hpi, 12hpi (K'-K''') and 1dpi (L'-L'''). Nuclei (white), eGFP (green), p-S6 (magenta). (P) Quantification of the p-S6⁺ area in the RPE layer revealed that expression was low in the RPE of unablated larvae but increased significantly in ablated larvae from 3hpi to 3dpi. p-values: ** ≤ 0.01, *** ≤ 0.001. Statistical information can be found in S9 Table. Dorsal is up and distal is left. Scale bar = 50μm.

after 12h in UI eyes (Fig 1B–1H). This low level of p-S6 staining in UI eyes likely results from 1-phenyl-2-thiourea (PTU) treatment, a thyroid hormone synthesis inhibitor commonly used to inhibit melanin synthesis in zebrafish and part of the RPE ablation paradigm (see Materials and Methods) (S1 Fig). PTU has been reported to have a modest effect on eye development in zebrafish [40,41] and our data indicate that early exposures result in low-level mTOR pathway activation. Likewise, in MTZ-treated wide-type larvae, the p-S6 signal was almost undetectable indicating that MTZ treatment does not elicit mTOR activation in the absence of RPE injury (S2 Fig). By contrast, p-S6 levels were significantly increased in the RPE layer of ablated (MTZ⁺) larvae from 6hpi to 3dpi, with peak activation detected during 6-12hpi (Fig 1I–1M and 1P). From 6hpi to 1dpi, p-S6 colocalized with eGFP in RPE cells, confirming that the mTOR pathway was activated in the RPE after ablation (Fig 1J–1L). However, at 2 and 3dpi, while p-S6 remained enriched in eGFP⁺ RPE, some p-S6⁺ cells also occupied the central region where the ablated RPE had been cleared (Fig 1M and 1N), suggesting that mTOR could also be active in regenerating RPE and/or cells that infiltrated the RPE layer during the response to injury. In addition to the RPE, p-S6 signal was also observed in the retinae of MTZ-treated larvae (e.g. Fig 1I–1M; arrowheads). Given that RPE ablation leads to rapid degeneration of the photoreceptors [18], these p-S6⁺ cells are likely Müller glia that have initiated a regenerative response to repair the retinal damage [42], a process known to involve mTOR activation [37,38].

## mTOR activity is required for RPE regeneration

To determine whether mTOR activity was required during RPE regeneration, we utilized complementary pharmacological and genetic perturbations to mTOR and assessed the regenerative ability of the RPE. To pharmacologically inhibit mTOR activity, rapamycin, an allosteric inhibitor of mTORC1 [43,44], and INK128, an adenosine triphosphate competitive inhibitor targeting both mTORC1 and mTORC2 [26], were utilized. Larvae were immersed in system water containing 2μM rapamycin, 0.9μM INK128, or DMSO as a vehicle control, from 24 hours prior to RPE ablation until they were sacrificed for analyses (Fig 2F). p-S6 distribution in the RPE at 2dpi was completely abolished by rapamycin or INK128 treatment (Fig 2A–2E), demonstrating the efficacy of these treatment paradigms in blocking mTOR activity in zebrafish. *rpe65a*:nfsB-eGFP⁺ cell morphology was visibly disrupted in MTZ⁺ larvae (S3A', S3B', S3D', and S3E' Fig) when compared with unablated MTZ⁻ controls (S3F–S3J Fig). The addition of mTOR inhibitors did not affect the magnitude of MTZ-mediated RPE ablation, as TUNEL staining showed comparable TUNEL⁺ puncta between rapamycin- and INK128-treated larvae and DMSO controls both in photoreceptors/RPE (S3A, S3B, S3D, S3E, and S3K Fig) and in RPE alone (S3A", S3B", S3D", S3E", and S3L Fig).

  To evaluate the effects of mTOR inhibition on RPE regeneration, we first assessed RPE proliferation post-ablation, which has been used previously as a quantitative readout of the regenerative response [18,19]. We performed 24-hour BrdU incorporation assays and assessed pigment recovery in rapamycin- or INK128-treated and DMSO-treated MTZ⁺ and MTZ⁻ larvae at 4dpi, the time at which proliferation within RPE layer peaks post-injury[18]. Quantification of the number of BrdU⁺ cells within the RPE layer showed that rapamycin and INK128 treatments significantly decreased proliferation post-injury, when compared to 4dpi DMSO-treated controls (Fig 2L–2P). There were no obvious differences between rapamycin- or INK128-treated or DMSO-treated MTZ⁻ larvae (Fig 2G–2K). Next, we quantified RPE pigment recovery as a readout of the extent to which the RPE had regenerated [18,19]. Consistent with BrdU incorporation results, RPE pigment recovery was also significantly diminished in rapamycin- and INK128-treated larvae when compared to DMSO-treated controls

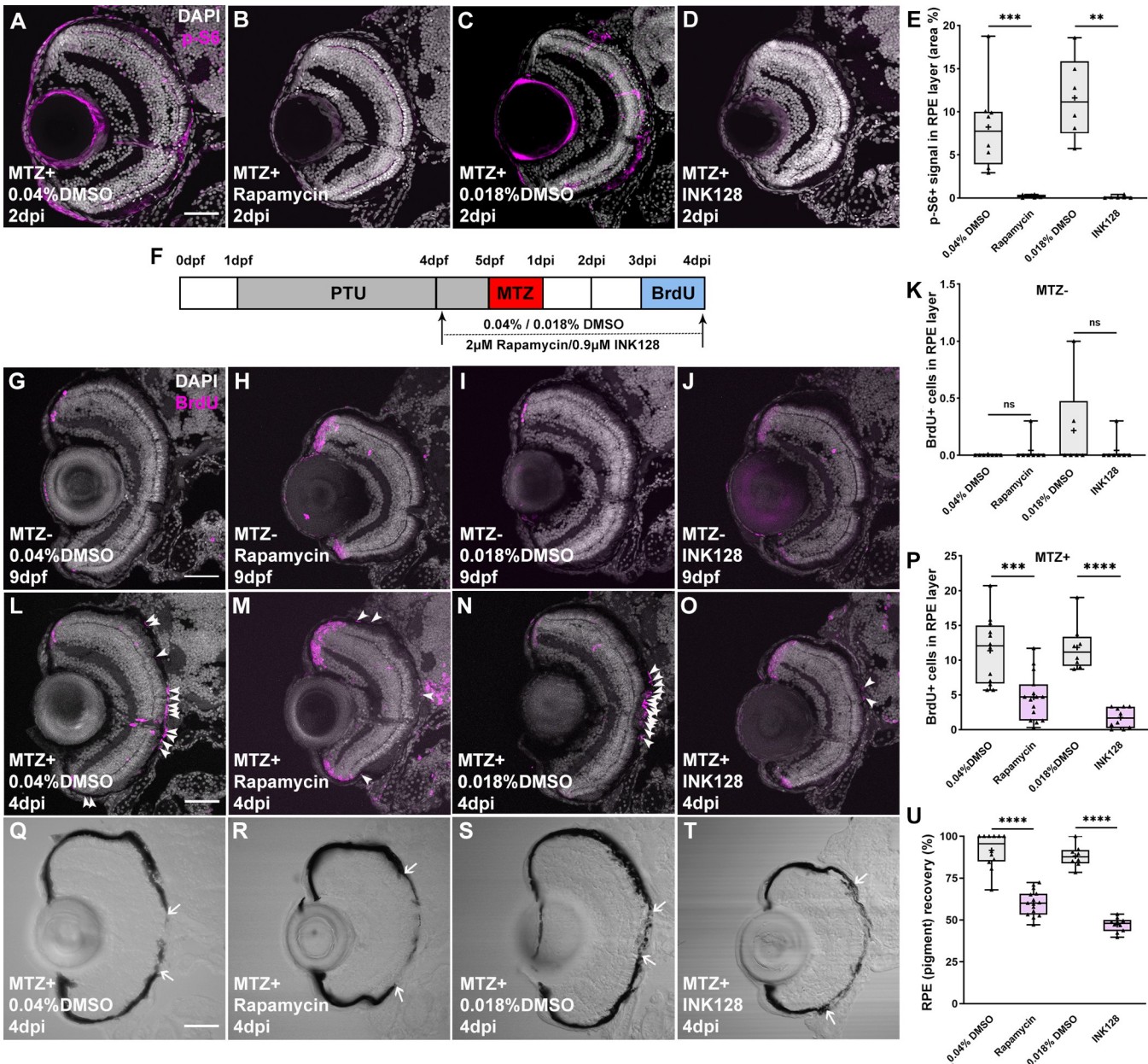

**Fig 2. Pharmacological inhibition of mTOR activity impairs RPE regeneration.** (A-D) Immunofluorescent images of p-S6 staining on transverse cryosections from MTZ+ DMSO-, rapamycin- or INK128-treated larvae at 2dpi. Nuclei (white), p-S6 (magenta) (E) p-S6 levels in the RPE layer were significantly decreased in rapamycin-treated and INK128-treated larvae when compared with DMSO-treated controls. (F) Schematic of the experimental paradigm showing the timeline for chemical treatments and ablation. (G-J) Immunofluorescent images of BrdU staining on transverse cryosections from MTZ- DMSO-, rapamycin- and INK128-treated larvae at 9dpf and (L-O) from MTZ+ DMSO-, rapamycin- or INK128 -treated larvae at 4dpi. *White arrowheads* indicate BrdU+ cells in the RPE layer. Nuclei (white), BrdU (magenta). (K) Quantification of BrdU+ cells in the MTZ- RPE layer showed no significant differences between DMSO-treated and inhibitor-treated larvae. (P) Quantification of BrdU+ cells in the RPE layer showed significantly fewer BrdU+ cells in MTZ+ rapamycin- and INK128-treated larvae when compared to DMSO-treated controls. (Q-T) Brightfield images of cryosections from MTZ+ DMSO-, rapamycin- or INK128-treated larvae at 4dpi. *White arrows* indicate the edges of pigment recovery. (U) Quantification of percent RPE recovery showed a significant impairment in pigment recovery in rapamycin- or INK128-treated larvae. p-values: ** $\leq$ 0.01, *** $\leq$ 0.001, and **** $\leq$ 0.0001. Statistical information can be found in S9 Table. Dorsal is up and distal is left. Scale bar = 50μm.

(Fig 2Q–2U). Rapamycin and INK128 showed no effect on RPE pigmentation in MTZ- controls (S4A–S4D Fig). We did note that rapamycin- and INK128-treated larvae possessed smaller eyes when compared to DMSO-treated siblings, and this was unrelated to MTZ

treatment (Fig 2G–2O). To assess this, we quantified retinal size in short-term (4-5dpf) and long-term (4dpf-4dpi) inhibitor treated larvae. Short-term treatment did not affect retinal size while long-term treatments resulted in a significant decrease in overall retinal size (S5 Fig). mTOR is known to modulate stem and progenitor cell behaviors in a number of tissues (e.g. [45,46]), including in hematopoietic stem cells, where mTOR inhibition results in a loss of quiescence and increased proliferation [46], and in the retinal ciliary marginal zone (CMZ) of *Xenopus*, where mTOR inhibition prevents CMZ-derived progenitor cell differentiation [47]. Interestingly, we detected a significant increase in proliferation in the CMZ of rapamycin- and INK128-treated larvae when compared to DMSO-treated siblings, which was unrelated to MTZ treatment (S6 Fig). Thus, in the zebrafish retina, it is possible that long-term mTOR inhibition also leads to attenuation of CMZ-derived retinal progenitor cell differentiation, while increasing retinal stem cell self-renewal.

Pharmacological inhibition of mTOR resulted in impaired RPE regeneration, supporting a role for mTOR activity in the regenerative response. We next wanted to confirm these data genetically; here, we took advantage of a zebrafish *mtor* mutant allele (*mtor^sa16755^*), which contains a nonsense mutation that is predicted to truncate the mTOR protein by 64% [48], and crossed the *rpe65a*:nfsB-eGFP ablation transgene into this mutant background. Previous studies showed that *Mtor* knockout in mouse and *Drosophila* (*dtor*) resulted in embryonic lethality [49,50], and in a different zebrafish *mtor* mutant allele (*mtor^xu015^*), most homozygous *mtor* mutants died by 10dpf [51]. In our assays, *rpe65a*:nfsB-eGFP; *mtor^sa16755^* larvae (hereafter referred to as *mtor^-/-^*) survived to at least 8dpf, with some *mtor^-/-^* larvae even inflating their swim bladders. However, at 9dpf, roughly half of the homozygous mutants had died, with the remainder appearing generally healthy. Therefore, our analyses of RPE regeneration were necessarily limited to these 9dpf/4dpi *mtor^-/-^* surviving larvae, which could have lower expressivity of *mtor* loss-of-function phenotypes. Importantly, mTOR activity after MTZ-mediated ablation, as assessed by p-S6 immunostaining at 2dpi, was significantly reduced in *mtor^-/-^* larvae, validating the utility of the *mtor^sa16755^* allele (Fig 3A–3C). As with the pharmacological perturbations, 24-hour BrdU incorporation from 3-4dpi and subsequent immunohistochemistry revealed significant decreases in proliferative cells within the RPE layer of *mtor^-/-^* larvae, when compared to *mtor^+/+^* controls (Fig 3F–3H). There were no significant differences in BrdU incorporation between *mtor^+/+^* and *mtor^-/-^* MTZ⁻ controls (Fig 3D, 3E, and 3H). Finally, RPE regeneration was also significantly impaired in *mtor^-/-^* larvae, as measured by both eGFP recovery and ZPR2 expression (Fig 3I–3J). These findings confirm the results from pharmacological inhibition of mTOR activity (Fig 2) and, collectively, provide three independent perturbations that support a requirement for mTOR activity in facilitating RPE regeneration.

## Activation of mTOR signaling enhances RPE regeneration

To determine whether activating mTOR signaling can enhance RPE regeneration, we utilized an mTOR agonist, MHY1485, to activate the pathway after RPE ablation [52]. Larvae were placed into either DMSO- or 2μM MHY1485-containing system water from 24 hours prior to RPE ablation until they were sacrificed for analyses (Fig 4A). To confirm the efficacy of MHY1485-mediated mTOR activation, p-S6 levels in the RPE layer were quantified at 2dpi and shown to be significantly increased when compared to MTZ⁺ DMSO-treated controls (Fig 4B, 4C, and 4R). To evaluate the effect of MHY1485 on proliferation within the RPE layer post-injury, we quantified the number of BrdU⁺ cells within the RPE layer at 2, 3, and 4dpi. While there were no significant differences between MTZ⁺ MHY1485-treated and DMSO-treated groups at 2dpi (Fig 4F,4G, and 4T) or 4dpi (Fig 4N,4O and 4T), at 3dpi, MTZ⁺ MHY1485-treated larvae possessed significantly more BrdU⁺ cells in the RPE layer when

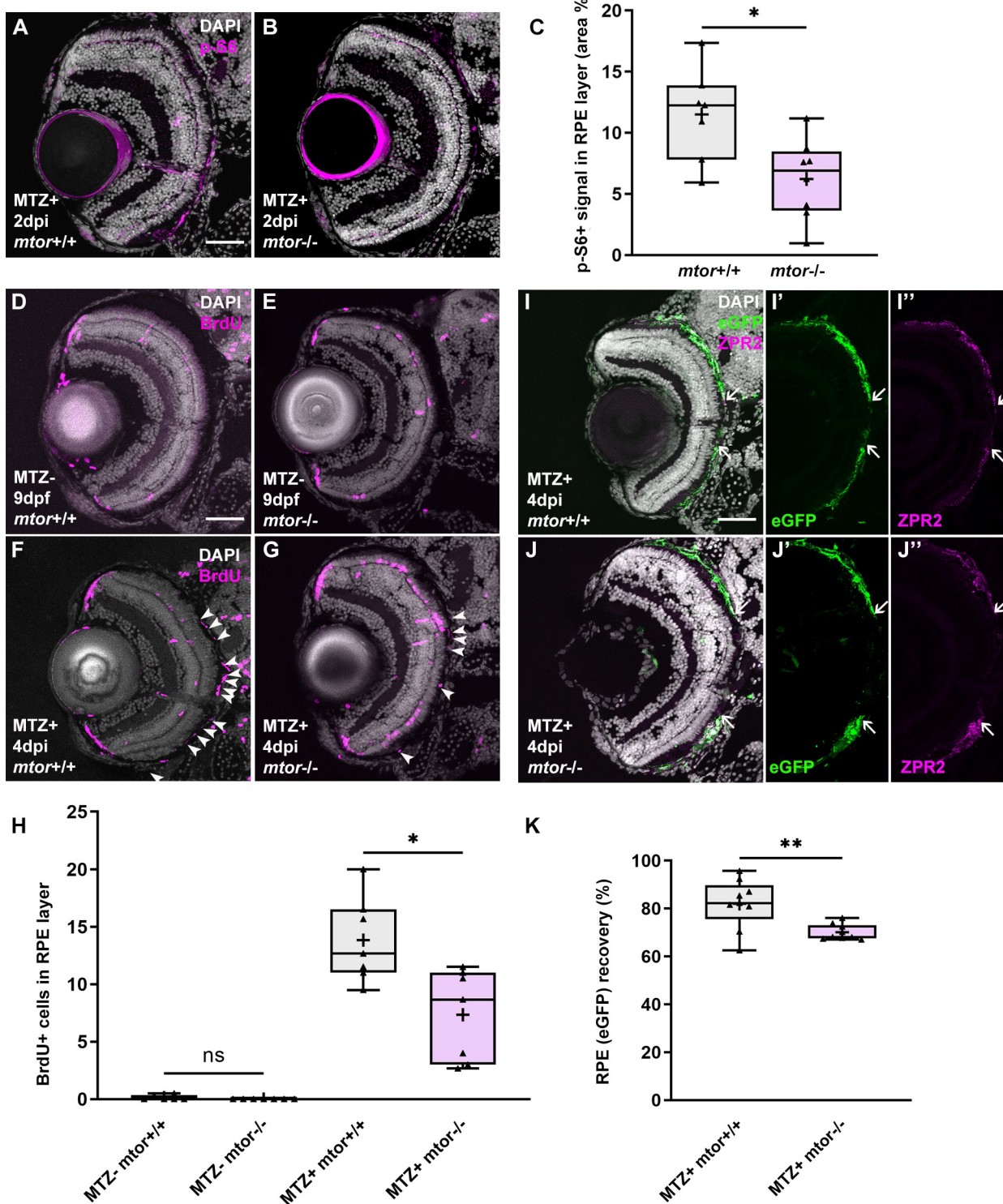

**Fig 3. Mutation in *mtor* impairs RPE regeneration.** (A,B) Immunofluorescent images of p-S6 staining on transverse cryosections from ablated *mtor*+/+ and *mtor*-/- at 2dpi. Nuclei (white), p-S6 (magenta). (C) Quantification of the p-S6 signal in the RPE layer revealed a significant decrease in MTZ+ *mtor*-/- larvae, when compared to MTZ+ *mtor*+/+ siblings. (D,E) Immunofluorescent images of BrdU staining on transverse cryosections from MTZ- (D,E) and MTZ+ (F,G) *mtor*+/+ and *mtor*-/- larvae at 9dpf/4dpi Nuclei (white), BrdU (magenta). (H) Quantification of BrdU+ cells in the RPE layer showed no significant differences between MTZ- *mtor*+/+ and *mtor*-/- larvae, but a significant decrease in MTZ+ *mtor*-/- larvae when compared to MTZ+ *mtor*+/+ siblings. (I-J) Immunofluorescent images of ZPR2 staining on transverse cryosections from MTZ+ *mtor*+/+ and *mtor*-/- larvae at 4dpi. Single channel (I' J') *rpe65a*:nfsB-eGFP images and (I" J") ZPR2 images. Nuclei (white), eGFP (green), ZPR2 (magenta). *White*

*arrows* highlight the edges of eGFP or ZPR2 recovery in the RPE. (K) Quantification of the eGFP expression recovery showed a significant decrease in MTZ$^+$ *mtor*$^{-/-}$ larvae, compared to *mtor*$^{+/+}$siblings. p-values: * $\leq$ 0.05, ** $\leq$ 0.01. Statistical information can be found in S9 Table. Dorsal is up and distal is left. Scale bar = 50μm.

compared with DMSO-treated siblings (Fig 4J,4K, and 4T). Indeed, MHY1485 treatment pushed forward the peak of proliferation in the RPE post-ablation to 2-3dpi, one day earlier than in normal regenerating conditions [18]. Notably, MHY1485 showed no impact on proliferation in unablated larvae (Fig 4D, 4E, 4H, 4I, 4L, 4M, and 4T). Given this increase in proliferation within the RPE layer after MHY1485 treatment at 3dpi, we next wanted to determine the effect of MHY1485 treatment on RPE regeneration. Here, we quantified central ZPR2 recovery at 3dpi and detected a significant increase in overall regeneration in the MHY1485-treated larvae when compared to DMSO controls (Fig 4P, 4Q, and 4S). Importantly, MHY1485 treatment did not impair MTZ-mediated ablation, as shown by TUNEL staining (S3A, S3C, S3K, and S3L Fig). When combined with our loss-of-function studies, these data support a model in which mTOR activity is a key regulator of RPE regeneration in zebrafish.

## Immune-related gene expression is downregulated when mTOR activity is impaired during RPE during regeneration

We next sought to identify possible mechanisms by which mTOR regulates RPE regeneration. To achieve this, we performed RNA-seq analyses on fluorescence-activated cell sorting (FACS)-isolated eGFP$^+$ RPE cells from MTZ$^-$ and MTZ$^+$ DMSO-treated and rapamycin-treated *rpe65a*:nfsB-eGFP larvae at 2dpi/7dpf and 4dpi/9dpf (Fig 5A and S1–S4 Tables). These time points represent early (2dpi) and peak (4dpi) regenerative stages [18]. Significantly differentially expressed genes (DEGs) were identified by comparing MTZ$^+$ DMSO- and rapamycin-treated expression profiles, with MTZ$^+$ DMSO replicates set as the control group. A total of 252 filtered DEGs were derived from the 2dpi dataset, among which 108 genes were upregulated and 144 were downregulated in rapamycin-treated RPE (Fig 5B and S1 and S2 Tables). In the 4dpi dataset, 948 filtered DEGs were identified, among which 195 genes were upregulated and 753 were downregulated in rapamycin-treated RPE (Fig 5B and S3 and S4 Tables). Relative to other cell type-specific gene markers, expression values of RPE-specific genes were high across the different treatment conditions examined by RNA-seq, supporting the enrichment of RPE in sorted eGFP$^+$ cell populations (S7 Fig). A small cohort of 2dpi/7dpf DEGs were verified by quantitative real-time PCR (qRT-PCR), providing technical validation of the dataset obtained from RNA-seq (S8 Fig). To identify mTOR-dependent mechanisms that could modulate RPE regeneration, we performed pathway enrichment analyses using STRING [53] and focused on downregulated DEGs in rapamycin-treated larvae. Interestingly, the most downregulated DEGs in rapamycin-treated larvae at 2dpi populated immune-related Reactome pathways (Fig 5C). DEGs within these immune-related pathways included proinflammatory cytokines (e.g. *il34*, *cxcl18a.1*), which are important signals for recruitment and chemotaxis of leukocytes [54,55], matrix remodeling proteins (e.g. *mmp9*, *qsox1*), which are crucial for the migration of RPE cells as well as macrophages [56–58], neutrophil regulation genes (e.g. *serpinb1*, *mpx*), amongst other genes associated with regulating an immune response (e.g. *iqgap2*, *itgb5*, *krt8*) (Fig 5D and S5 Table) [59–61]. Importantly, most of these immune-related DEGs were upregulated in MTZ$^+$ DMSO-treated controls, when compared to MTZ$^-$ DMSO-treated controls (Fig 5D and S6 Table), consistent with our previous study [19]. Taken together, these data support a model in which immune system activity is upregulated in RPE cells at an early stage of regeneration (2dpi) in an mTOR-dependent manner.

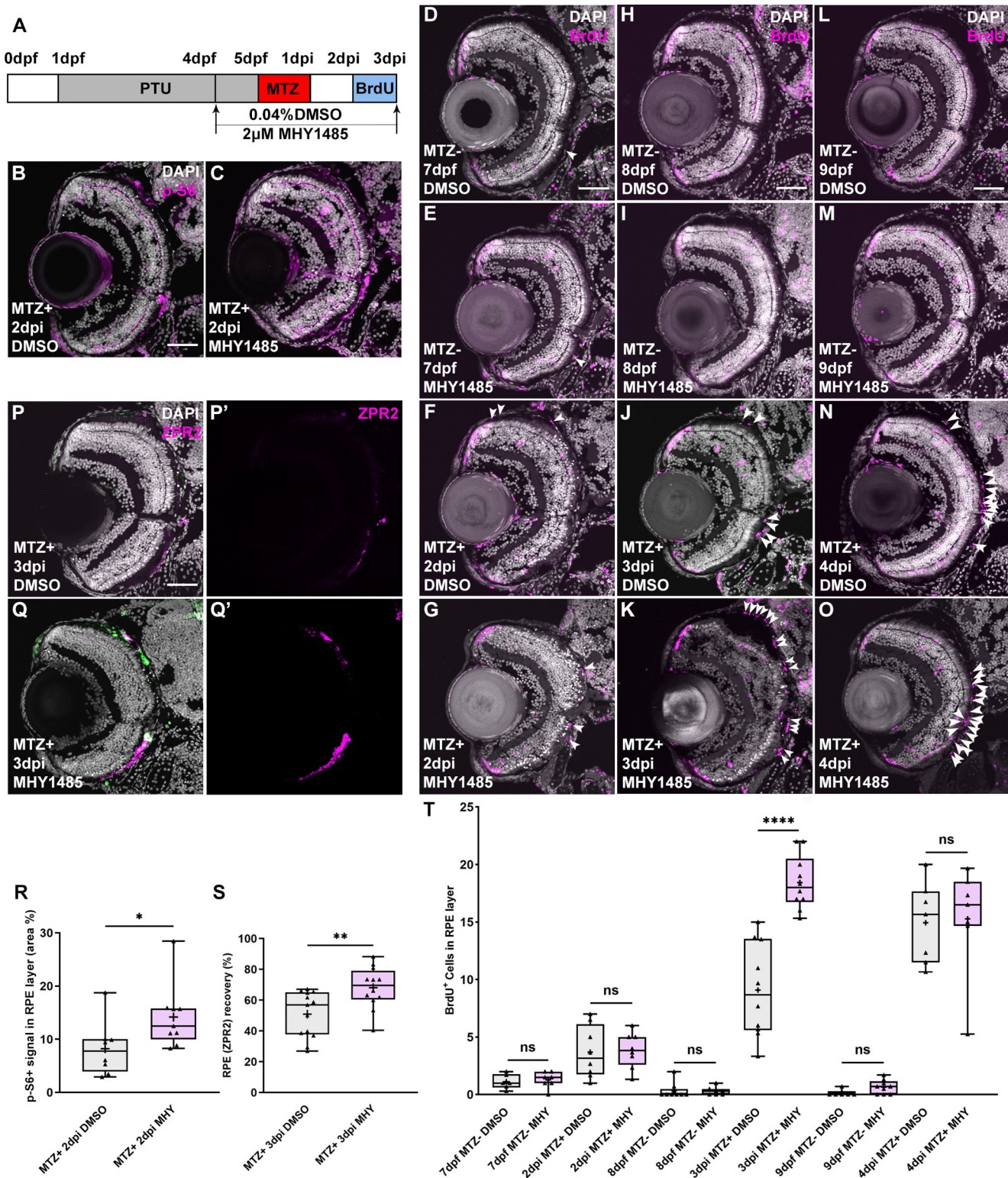

**Fig 4. Pharmacological activation of mTOR signaling enhances RPE regeneration.** (A) Schematic of the experimental paradigm showing the timeline for chemical treatments and ablation. (B,C) Immunofluorescent images of p-S6 staining on transverse cryosections from MTZ[+] DMSO- and MHY1485-treated larvae at 2dpi. Nuclei (white), p-S6 (magenta). (R) Quantification of p-S6 signal in the RPE layer showed a significant increase of p-S6 levels in MHY1485-treated larvae when compared to DMSO-treated controls. (D-O) Fluorescent images of BrdU immunostaining on cryosections from MTZ[-] and MTZ[+] DMSO- and MHY1485-treated larvae at (D-G) 7dpf/2dpi, (H-K) 8dpf/3dpi and (L-O) 9dpf/4dpi. *White arrowheads* highlight BrdU[+] cells in the RPE

layer. Nuclei (white), BrdU (magenta). (T) Quantification of BrdU$^+$ cells in the RPE layer revealed significantly increased proliferative cells in the RPE of MTZ$^+$ MHY1485-treated larvae when compared to DMSO-treated controls at 3dpi, but no significant differences between ablated MHY1485-treated larvae and DMSO-treated siblings at 2dpi or 4dpi. There were no significant effects of MHY1485 treatment on MTZ$^-$ larvae across all time points. (P-Q) Fluorescent images of ZPR2 immunostaining on cryosections from MTZ$^+$ DMSO- and MHY1485-treated larvae at 3dpi. *White arrows* highlight the edges of ZPR2 signal recovery. Nuclei (white), eGFP (green), ZPR2 (magenta). (S) Quantification of percent ZPR2 recovery showed a significant increase in MHY1485-treated larvae over DMSO-treated controls. p-values: * $\leq$ 0.05, ** $\leq$ 0.01, **** $\leq$ 0.0001. Statistical information can be found in S9 Table. Dorsal is up and distal is left. Scale bar = 50μm.

At 4dpi, immune-related pathways were no longer enriched in the downregulated 4dpi dataset when comparing rapamycin- to DMSO-treated controls. Downregulated genes at this time point comprised pathways related to small molecule transport and cell-cell communication pathways, including GABA, neurotransmitter, and G-protein signaling (Fig 5C and S7 Table). Taken together, these data suggest a model in which mTOR acts early during regeneration as a regulator of the immune response, while later, mTOR-dependent gene regulation perhaps facilitates some of the restoration/reintegration of regenerated RPE cells into a functional tissue.

## mTOR signaling regulates the recruitment of macrophages/microglia during RPE regeneration

Recently, we showed that macrophages/microglia infiltrate the injury site post-RPE ablation and are important regulators of zebrafish RPE regeneration [19]. That rapamycin impaired the expression of immune-related genes and pathways suggests that mTOR activity may be required for leukocyte recruitment and/or function during RPE regeneration. To test this hypothesis, we utilized *mpeg1*:mCherry;*rpe65a*:nfsB-eGFP transgenic fish, in which macrophages and microglia are labeled by mCherry [62]. MTZ$^+$ and MTZ$^-$ control *mpeg1*:mCherry; *rpe65a*:nfsB-eGFP larvae were treated with rapamycin or DMSO from 4dpf until 3dpi (Fig 6A), the time at which macrophage/microglia infiltration peaks after RPE ablation [19]. Immunostaining for mCherry revealed no visible accumulation of macrophages/microglia in the RPE layer of DMSO-treated MTZ$^-$ larvae (Fig 6B), while, as expected, significant macrophage/microglia recruitment to the RPE layer in DMSO-treated MTZ+ larvae was observed (Fig 6D and 6F). By contrast, macrophage/microglia presence in the RPE layer was significantly reduced in MTZ$^+$ rapamycin-treated larvae (Fig 6D–6F). These data demonstrate that mTOR activity is required for leukocyte recruitment to the injured/regenerating RPE.

## Macrophage/microglia function is required to maintain mTOR activity in the RPE during regeneration in an inflammation-independent manner

mTOR activation within the RPE layer peaks between 6-12hpi (Fig 1) and mTOR activity is required for the recruitment of macrophages/microglia to the RPE post-injury (Fig 6). However, significant macrophage/microglia recruitment to the injured RPE does not occur until 2dpi, and peaks at 3dpi [19], which suggests a model in which early mTOR activation in the RPE contributes to the recruitment of macrophages/microglia to the injury site. As mTOR activity remains stably elevated in the regenerating RPE until 3dpi, and cytokine signaling is a well-known extrinsic inducer of mTOR activity in a variety of cell types [63,64], we hypothesized that there may be crosstalk between recruited macrophages/microglia and the RPE, whereby macrophages/microglia could also play a role in maintaining mTOR activation in the RPE at later stages post-injury. Indeed, during retinal regeneration in zebrafish, macrophage/ microglia-mediated inflammation promotes mTOR activation in Müller glia and mTOR activity is required for regeneration [37]. To test this hypothesis and explore the relationship

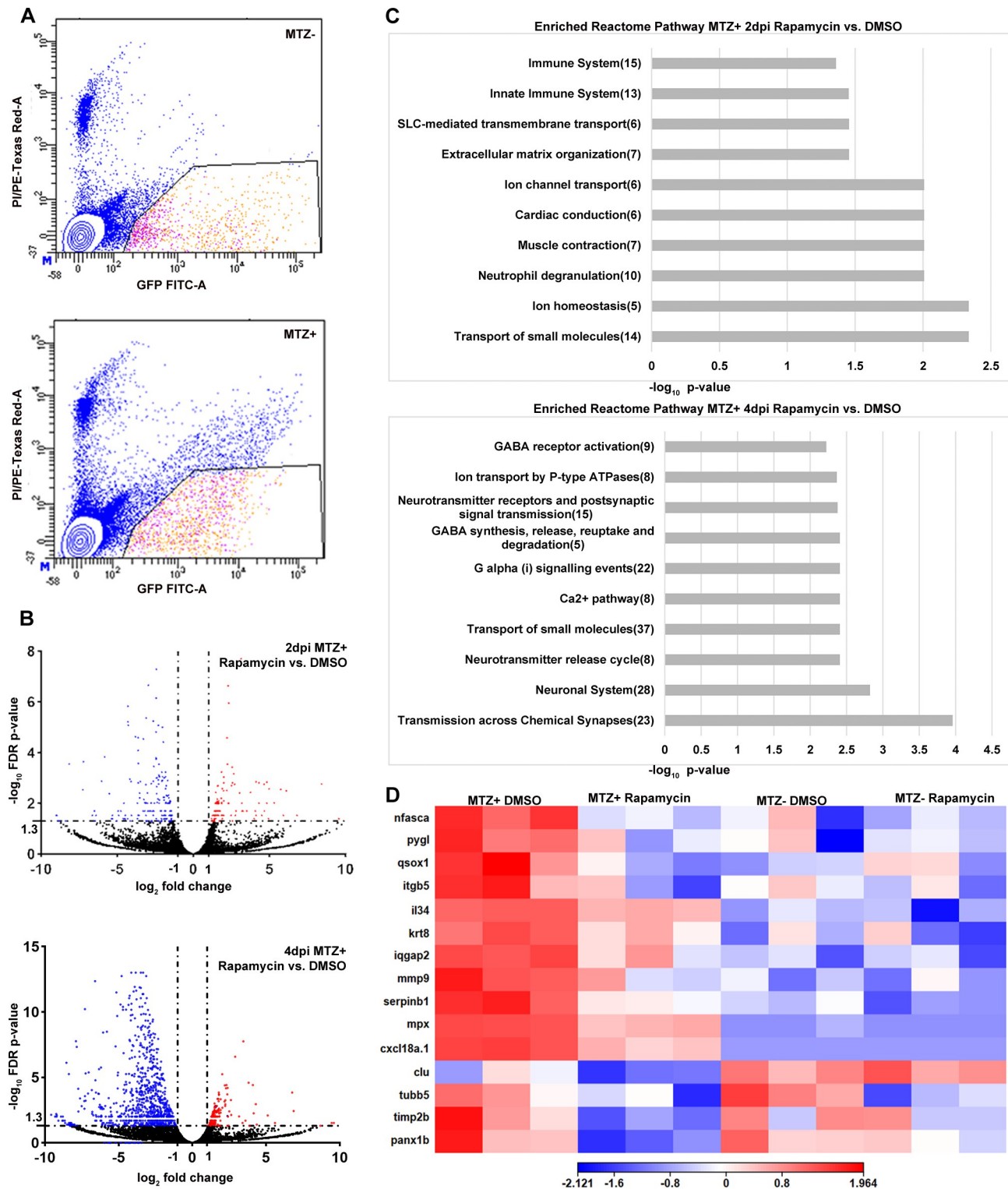

**Fig 5. Inhibition of mTOR signaling affects the expression of immune related genes in regenerating RPE cells.** (A) Representative FACS plots from MTZ⁻ and MTZ⁺ samples to show gating parameters used to isolate GFP⁺ PI⁻ RPE cells (B) Volcano plots showing differentially expressed genes between MTZ⁺ rapamycin-treated and DMSO-treated groups at 2dpi and 4dpi. Dashed lines indicate the threshold criteria. Each dot represents an individual gene; red dots represent significantly upregulated DEGs, blue dots represent significantly downregulated DEGs, and black dots represent non-significant genes. (C) Pathway enrichment analysis on the significantly downregulated 2dpi and 4dpi DEGs. Numbers in parentheses are the gene counts enriched in each pathway. (D) Hierarchical clustering heatmap of the 15 genes enriched in the 2dpi immune system Reactome pathway across all four

experimental groups showing these immune related genes are downregulated by rapamycin treatment. Heatmap legend represents $\log_{10}$ (counts per million mapped reads; CPM).

between macrophages/microglia and later mTOR activity in the RPE, we used the CSF1R inhibitor, pexidartinib (PLX3397), to deplete macrophages/microglia, an approach that has been shown to be effective in zebrafish [19,65–67], and assessed the effects on mTOR activation. Importantly, the *csf1ra* and *csf1rb* genes were not significantly upregulated in the MTZ[+] RPE at 2dpi and therefore unlikely to be directly involved in the regenerative response of RPE cells at this time (S8 Table). Experimentally, larvae were exposed to 1μM PLX3397 or DMSO from 72 hours prior to ablation until 2dpi (Fig 7A), the time when significant recruitment of macrophages/microglia to the RPE post-injury was first observed [19]. As expected, MTZ[+] larvae exposed to PLX3397 showed a significant reduction in *mpeg1*-driven mCherry signal in the RPE when compared to DMSO-treated controls (Fig 7B–7E and 7H). Having confirmed the efficacy of PLX3397 in depleting macrophages/microglia, we next sought to determine whether macrophage/microglia depletion affected mTOR activation in the RPE post-injury. p-S6 levels in the RPE were significantly suppressed by PLX3397 treatment at 2dpi, when compared to DMSO controls (Fig 7B, 7C, 7F, 7G, and 7I). These data demonstrate that RPE damage induces an mTOR-dependent recruitment of macrophages/microglia to the injury site and macrophages/microglia reinforce mTOR activity in the regenerating RPE. Interestingly, mTOR signaling also appeared to be activated in macrophages/microglia at 2dpi, as shown in

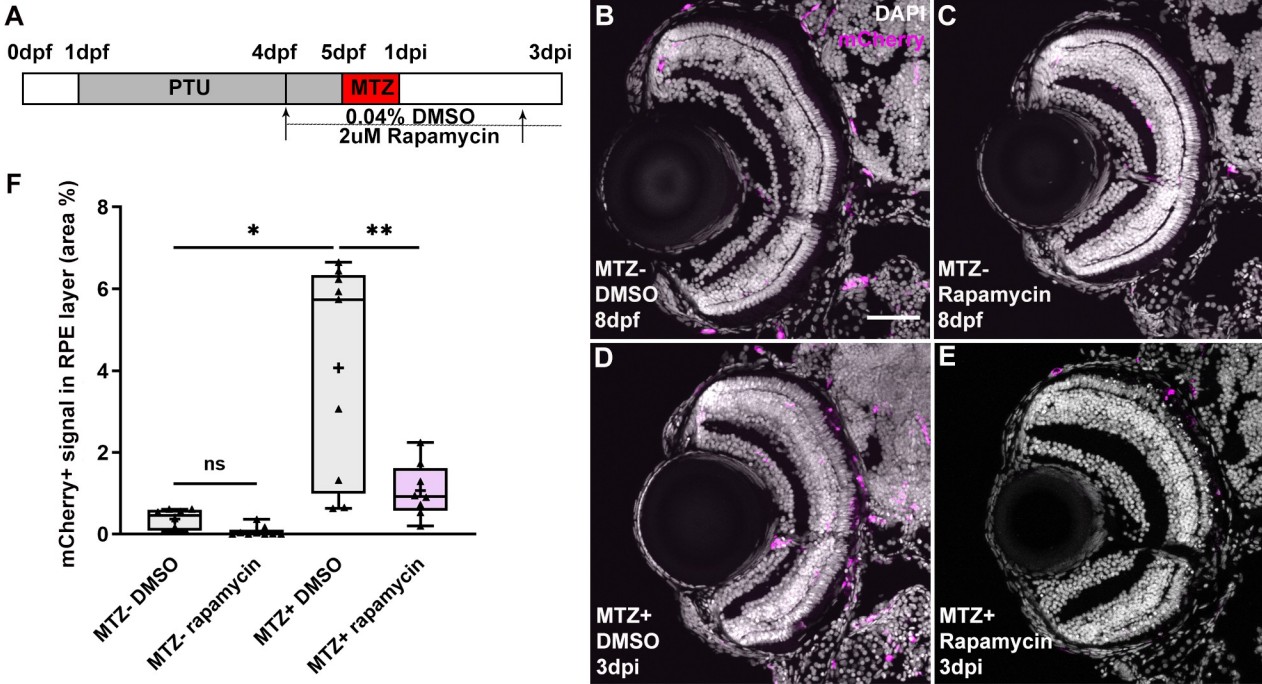

**Fig 6. Inhibition of mTOR activity impaired macrophage/microglia recruitment to the RPE layer post-ablation.** (A) Schematic of the experimental paradigm showing the timeline for chemical treatments and ablation on transgenic larvae (*mpeg1*:mCherry;*rpe65a*:nfsB-eGFP). (B-E) Immunofluorescent images of mCherry staining on transverse cryosections from MTZ[-] and MTZ[+] DMSO- and rapamycin-treated larvae at 3dpi. Nuclei (white), mCherry (magenta). (F) Quantification of the mCherry signal in the RPE layer showed a significant increase in MTZ[+] DMSO-treated larvae, when compared to MTZ[-] DMSO controls. The percent of the RPE covered by mCherry[+] cells was significantly reduced in MTZ[+] rapamycin-treated larvae when compared to MTZ[+] DMSO-treated controls. However, there was no significant difference between MTZ[-] DMSO-treated and rapamycin-treated groups. p-values: * ≤ 0.05, ** ≤ 0.01. Statistical information can be found in S9 Table. Dorsal is up and distal is left. Scale bar = 50μm.

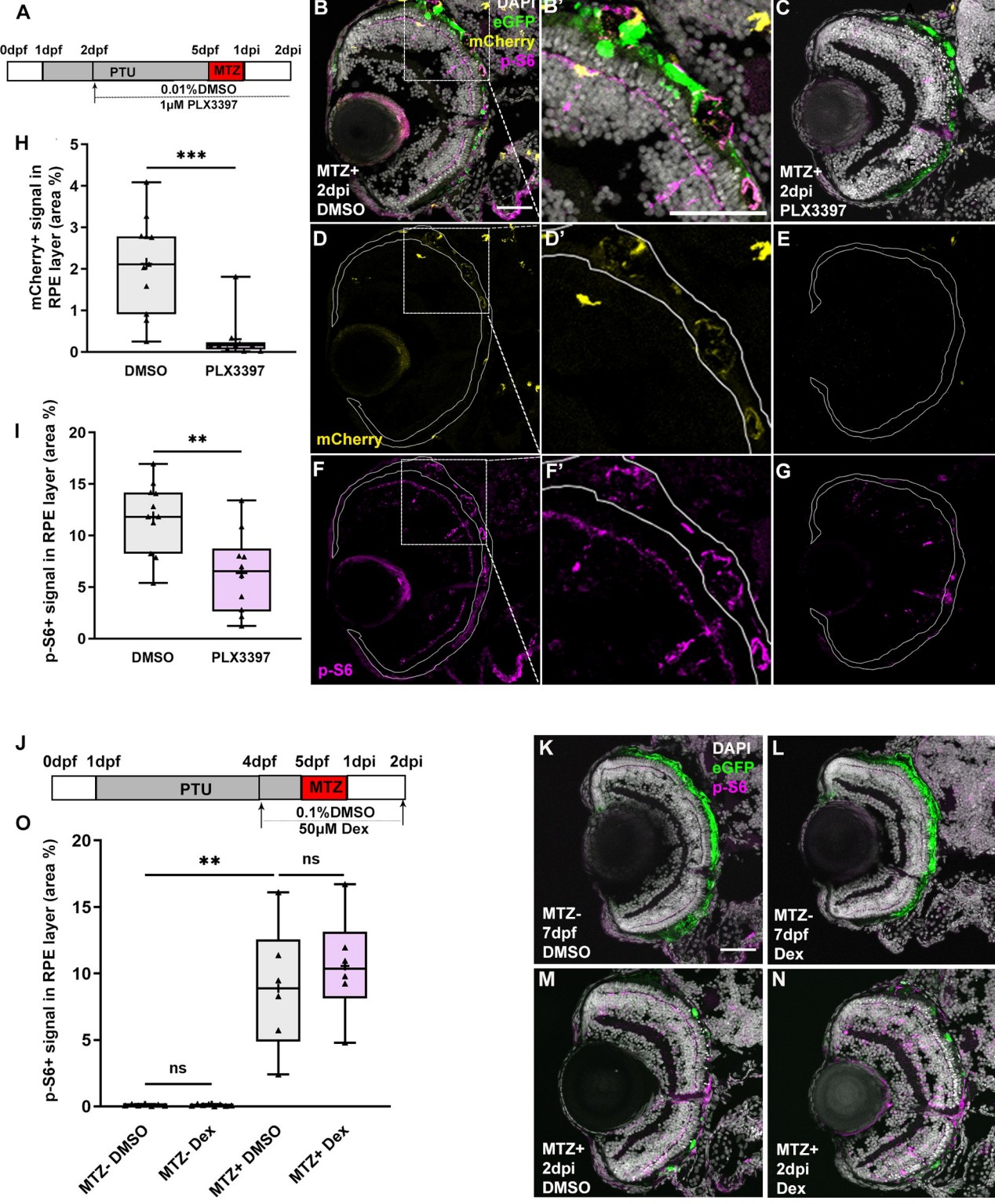

**Fig 7. Macrophage/microglia function is required for the maintenance of mTOR activity in an inflammation-independent manner.** (A,J) Schematic of the experimental paradigm showing the timeline for chemical treatments and ablation on transgenic larvae (*mpeg1*:mCherry;*rpe65a*:nfsB-eGFP). (B-G) MTZ+ DMSO- and PLX3397-treated larvae at 2dpi. Single channel immunofluorescent images of mCherry (D,E) and p-S6 (F,G) are shown. (B',D',F') are high-magnification images showing the colocalization of p-S6 and mCherry. Nuclei (white), eGFP (green), p-S6 (magenta), mCherry (yellow). (H) Quantification of mCherry signal in the RPE layer showed a significant depletion of mCherry+ macrophages/microglia in the

RPE after PLX3397 treatment in MTZ[+] larvae, compared to DMSO-treated controls. (I) Quantification of p-S6 levels in the RPE layer showed a significant decrease in PLX3397-treated larvae, when compared to DMSO-treated controls. (K-N) Fluorescent images of p-S6 immunostaining on cryosections from MTZ[-] and MTZ[+] DMSO- and Dex-treated larvae at 7dpf/2dpi. Nuclei (white), eGFP (green), p-S6 (magenta). (O) Quantification of p-S6 level in the RPE layer showed a significant increase in MTZ[+] DMSO-treated larvae when compared to MTZ[-] DMSO-treated control, but no significant differences between Dex-treated and DMSO-treated larvae from MTZ[-] or MTZ[+] groups. p-values: ** $\leq$ 0.01, *** $\leq$ 0.001. Statistical information can be found in S9 Table. Dorsal is up and distal is left. Scale bar = 50μm.

MTZ[+] DMSO-treated controls (Fig 7B', 7D', and 7F'). As noted above, p-S6[+] cells were detected at 2dpi and 3dpi in the region where the ablated RPE had been cleared (Fig 1M and 1N). This observation suggests that the injury-dependent recruitment of macrophages/microglia to the ablated RPE also activates mTOR in the macrophages/microglia, perhaps modulating their behaviors during the injury/regenerative response.

In many tissue injury paradigms, the recruitment and activation of leukocytes triggers inflammation [68–71]. Inflammation is a complex process that, when acute and/or chronic, can be detrimental to tissue survival; if properly resolved, however, inflammation can stimulate pro-regenerative responses [72,73]. Indeed, inflammation is necessary for RPE and retinal regeneration in zebrafish [19,37,74]. With this in mind, we utilized Dexamethasone (Dex) to dampen inflammation systemically [19,74] to determine whether inflammation was responsible for mTOR activity in the RPE layer at later stages post-injury. As expected, DMSO-treated MTZ[+] larvae showed significantly increased p-S6 levels in the RPE layer at 2dpi when compared to MTZ[-] controls (Fig 7K, 7M, and 7O). Interestingly, there were no significant differences in mTOR activity between MTZ[+] Dex-treated and DMSO-treated larvae (Fig 7L, 7N, and 7O). Dex-treatment does not affect macrophage/microglia recruitment post-RPE ablation [19], and therefore these data support a model in which it is an inflammation-independent function of macrophages/microglia that maintains mTOR activity in the RPE layer during regeneration.

## Discussion

AMD is a progressive blinding disease characterized by dysfunction of the RPE and choroid which causes a gradual loss of these tissues, of the photoreceptors, and eventually of central vision [75]. Currently, there are few options for treating AMD [76]; however, an innovative potential direction for therapeutic development is the stimulation of intrinsic regeneration of the RPE [13]. Previously, our lab established a genetic RPE ablation paradigm in zebrafish and demonstrated that zebrafish are capable of regenerating a functional RPE monolayer, thus providing a model through which the molecular underpinnings of the RPE regenerative response can be identified [18].

In this study, we focused on activation of mTOR signaling, which has been associated with tissue and organ regeneration in a number of contexts and model systems [29,77]. mTOR activity was rapidly induced in zebrafish RPE cells within 6 hours after the onset of RPE ablation, and the mTOR pathway remained active through 3dpi, the time of peak proliferation within the regenerating RPE layer [18] (Fig 8A and 8B). Through a combination of pharmacological and genetic loss-of-function and gain-of-function assays, our results demonstrate that mTOR activity plays a critical role during RPE regeneration in zebrafish. Our experiments were performed in larval stage zebrafish and thus, it is possible that the role of mTOR is different in the adult RPE; however, we have shown that the regenerative response proceeds similarly between larvae and adult animals [18]. These results are consistent with findings in other systems, where the mTOR pathway has been shown to exert a pro-regenerative role. Indeed, in the zebrafish retina, mTOR is activated in Müller glia within 6 hours after injury and expression is maintained in proliferating Müller glia and Müller glia-derived progenitor cells

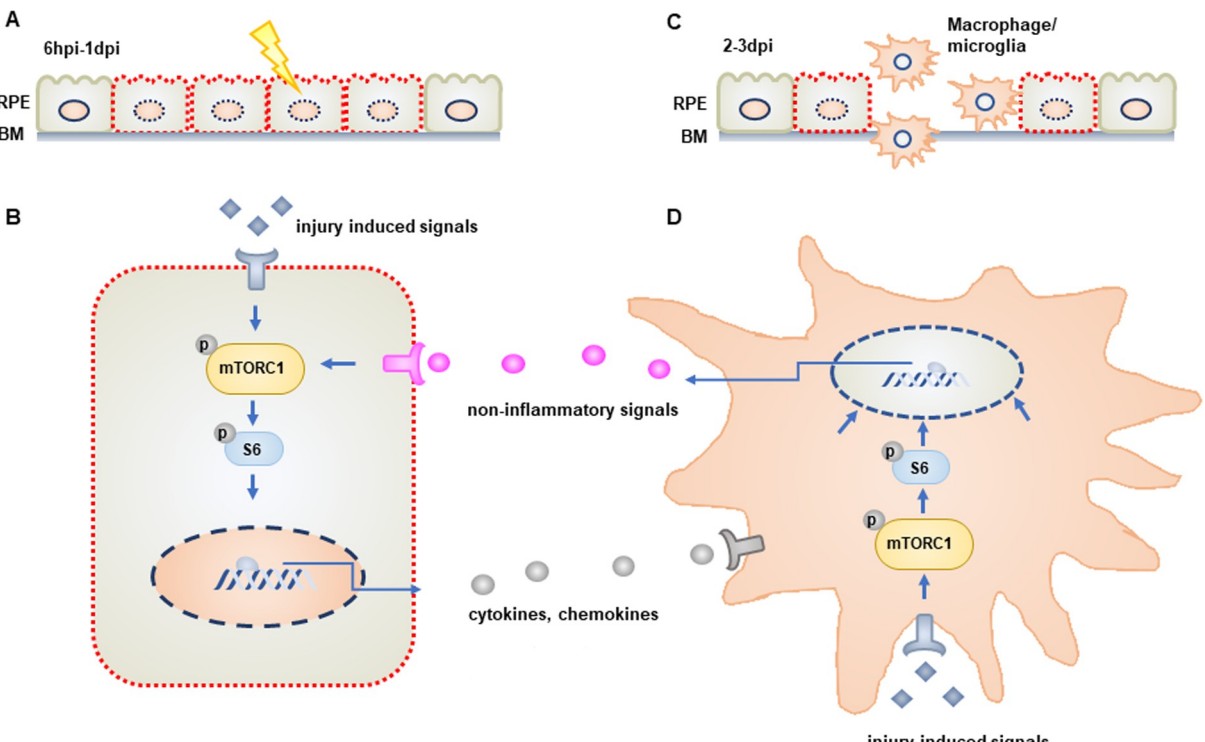

**Fig 8. Model of mTOR activity during RPE regeneration.** (A) mTOR activity is rapidly induced in damaged/regenerating RPE cells between 6hpi and 1dpi. (B) Activation of mTOR signaling in RPE cells regulates the expression of immune-related genes, including cytokines and chemokines. (C) mTOR activity is required for the recruitment of macrophages/microglia to the injured/regenerating RPE. (D) mTOR is also activated in macrophages/microglia recruited to the RPE post-injury and these cells are required to maintain mTOR activity in RPE cells during the later stages of the regenerative response. This maintenance occurs in an inflammation-independent manner.

(MGPCs) [37]. During retinal regeneration, mTOR activity is required both for the dedifferentiation of Müller glia to generate progenitors as well as the proliferation of the progenitor cells themselves. Similarly, in the chicken retina, mTOR is also required for the formation of MGPCs [38]. These results parallel our observations in the injured and regenerating RPE, as well as those in cultured RPE cells where mTOR activity is required for proliferation and migration of RPE cells after a laser photocoagulation injury [27]. Taken together, these data further support an evolutionarily conserved role for mTOR in modulating regenerative responses and extend this to include the RPE.

Interestingly, our results link mTOR activation in the RPE after injury to immune system activity (Fig 8C and 8D). Involvement of the immune system in regeneration is well documented in several tissues and organs [78], including the zebrafish RPE [19]. In RPE-ablated larvae where mTOR activity was inhibited, the recruitment of macrophages/microglia to the injured RPE was impaired. This suggests that mTOR-dependent regulation of gene expression in damaged RPE facilitates signaling events that recruit leukocytes to the injury site. Amongst the DEGs downregulated in the RPE of rapamycin-treated larvae, several genes involved in immune response activation were identified, and these include *il34* and *mmp9*. IL-34 is a proinflammatory cytokine that regulates macrophage differentiation, proliferation, migration, and polarization [54,79], and has been shown to be upregulated in the zebrafish RPE post-ablation [19]. Matrix metalloproteinase-9 (Mmp-9) is a secreted gelatinase that acts on extracellular molecules, such as extracellular matrix, growth factors, and cytokines [80,81]. Mmp-9 also plays a critical role in recruiting leukocytes to sites of injury after peripheral nerve crush in rats

[82], glomerulonephritis in mice [83], and cryoinjury in zebrafish [58]. mmp-9 also regulates heart regeneration in zebrafish by mediating leukocyte recruitment, possibly via the processing of chemokines into active forms [58]. Our data demonstrate that *il34* and *mmp9* are upregulated in an mTOR-dependent fashion, and these factors could stimulate macrophage/microglia recruitment to the injured RPE during the early stages of the injury response.

It is known that macrophages/microglia directly modulate the RPE regenerative response [19] and here, we demonstrate that these cells also reinforce mTOR activity in regenerating RPE cells during the later stages of regeneration. Interestingly, our results indicate that the macrophage/microglia-dependent maintenance of mTOR in the RPE is inflammation-independent. This is in contrast to mTOR activation during retinal regeneration, which is dependent on inflammation [37]. Cytokines have been shown to activate the mTOR pathway [64], and physiologically, macrophages are the main cellular source of many cytokines [84], including in zebrafish post-RPE injury [19]. Thus, it is possible that macrophages/microglia recruited to the RPE injury site release cytokines that function to maintain elevated mTOR activity in RPE cells as they regenerate. In this context, however, these cytokines would likely be of the anti-inflammatory subtype given that mTOR maintenance is inflammation-independent. Additionally, the mTOR pathway can be activated by a variety of intracellular and extracellular signals, the latter of which includes components of the insulin/insulin-like growth factor (IGF) pathway [85], Wnt pathway [86], and macrophage-derived glutamine [87]. Thus, it is also possible that one or more of these pathways could be stimulated by injury-induced macrophages/microglia and function to maintain mTOR activity in the regenerating RPE. Several zebrafish studies have identified injury-induced gene expression profiles in isolated leukocytes [19,88–90], and it will be of interest to screen candidates for potential mTOR maintenance roles during RPE regeneration. Finally, with respect to the immune system, we detected mTOR activity in RPE-localized macrophages/microglia at 2dpi (Fig 7B', 7D', and 7F'). Thus, we cannot rule out the possibility that genetic and pharmacological modulation of mTOR activity also affected the ability of macrophages/microglia to respond to RPE ablation. Indeed, mTOR activity in leukocytes is known to modulate their migratory abilities, supporting this notion [91]. Our current data do not enable us to differentiate between these possibilities and targeted loss-of-function assays in RPE-only and leukocytes-only will be needed to assess this.

Mechanistically, mTOR function during RPE regeneration may be bi-phasic, with an early stage that stimulates innate immune system activity and a later stage that directly regulates regeneration of a functional RPE monolayer. RNA-seq data from 4dpi rapamycin-treated samples identified genes/pathways that could function in RPE cells at later stages of the regenerative response. Importantly, we observed low relative expression values of retinal cell marker genes in sorted RPE cell populations across all samples, with the exception of some photoreceptor-specific genes (S7 Fig). RPE cells are interdigitated with photoreceptor outer segments in a process that facilitates their phagocytosis and the recycling of key components of the visual transduction cascade [2]. While we cannot rule out slight impurities in the sorted cell population, enrichment of photoreceptor genes could also reflect some inclusion in sorted RPE cells due to the close proximity of these cell types in the retina. Regardless, robust enrichment of RPE markers indicate high enrichment for RPE cells and thus, a useful dataset through which downstream mTOR-dependent pathways important for RPE regeneration can be further studied. For example, GABA signaling and neurotransmitter release pathways were significantly downregulated in the rapamycin-treated group. GABA has a well-known role in synaptic transmission in the central nervous system, including in the retina [92]. GABA receptors/transporters are expressed in cultured human RPE cells [93,94], bullfrog and mouse RPE [95,96], and chick RPE [97], where signaling regulates the intracellular calcium concentration. Calcium signaling plays a number of important roles during tissue regeneration [98] and

wound healing [99], and therefore mTOR-dependent activation of GABA signaling could serve as a trigger to induce migration of regenerating RPE cells and/or reformation of the RPE monolayer. GABA signaling has also recently been shown to be involved in tissue regeneration, including in the zebrafish retina [100,101], making this an interesting pathway for future studies during RPE regeneration. In addition to its role in leukocyte recruitment, mmp-9 has also been shown to be involved in regulating cell migration by remodeling the extracellular matrix [56,102,103]. Indeed, in cultured RPE cells, TNFα-induced MMP-9 expression promotes RPE cell migration, which is controlled by activation of Akt/mTORC1 signaling [56]. During RPE regeneration, mTOR activity could stimulate mmp-9-mediated modulation of RPE cell migration to enable the reformation of a functional RPE monolayer. Finally, mTOR is a critical mediator of cellular growth control via its ability to facilitate anabolism (protein synthesis) and inhibit catabolism (autophagy) in response to environmental inputs [21]. There is significant proliferation in the RPE layer during regeneration [18], which undoubtedly requires coordinated protein synthesis, cellular growth, and substantial energy requirements and it is likely that mTOR activity modulates these processes during RPE regeneration. Future studies focusing on these and other candidates will shed light on the mTOR-dependent processes that facilitate RPE regeneration in zebrafish and whether any of these processes can be stimulated in the mammalian RPE to trigger a regenerative response.

## Materials and methods

### Ethics statement

All experiments were performed with approval by the University of Pittsburgh School of Medicine Institutional Animal Care and Use Committee.

### Fish maintenance and husbandry

Adult zebrafish were maintained at 28.5˚C under a 14-hour light/10-hour dark cycle. Embryos and larvae used for subsequent experiments were kept in an incubator at 28.5˚C in the dark until being euthanized by tricaine (MS-222; Fisher Scientific).

### Genotyping

$mtor^{sa16755}$ [48] was purchased from the Zebrafish International Resource Center (ZIRC;). $mtor^{+/-}$ fish were crossed to: $rpe65a$:nfsB-eGFP fish[18] to generate $mtor^{+/-}$;$rpe65a$:nfsB-eGFP, which were identified by eGFP screening and genotyping via PCR-restriction fragment length polymorphism-based analysis (PCR-RFLP). $mtor^{+/-}$;$rpe65a$:nfsB-eGFP fishes were then incrossed to generate $mtor^{-/-}$;$rpe65a$:nfsB-eGFP larvae, which were genotyped by PCR-RFLP after euthanasia. For genotyping, genomic DNA was extracted from adult fins or larval tails by incubating with 50ul of 50mM NaOH at 95˚C for 10 minutes. After cooling to 4˚C, 5ul of 1M Tris-HCl (pH8) was added for neutralization and this mixture was used as a template for PCR amplification. Genotyping primers are included in S10 Table. For PCR-RFLP analysis, 10ul of PCR product was digested with 5 units of MseI (New England Biolabs, Ipswich, Massachusetts) for one hour. The digested PCR products were separated on 2% agarose gel. Uncut, 391bp fragments, indicated $mtor^{+/+}$; $rpe65a$:nfsB-eGFP and cut, 223 and 168 bp fragments, indicated $mtor^{-/-}$;$rpe65a$:nfsB-eGFP. Genotyping results from PCR-RFLP were confirmed by Sanger sequencing in a subset of larvae.

### RPE ablation

RPE ablation followed the protocol in Hanovice et al., (2019)[18]. Briefly, embryos derived from $rpe65a$:nfsB-eGFP crosses were treated with 1-phenyl-2-thiourea (PTU; Sigma-Aldrich)

from 6hpf to 5dpf to inhibit pigmentation. At 4dpf, larvae carrying the *rpe65a*:nfsB-eGFP transgene were selected by eGFP screening under a fluorescent microscope (Zeiss AxioZoom V.16). At 5dpf, larvae were treated with 10mM metronidazole (MTZ) in system water at times between 10am-2pm. 24 hours later, larvae were removed from MTZ and maintained in system water until euthanized for experiments.

## Pharmacological treatments

Drugs were diluted in system water and added 24 hours prior to MTZ treatment, to allow infiltration and effectiveness before RPE ablation, given the rapid elevation of mTOR activity in response to RPE damage (See Fig 1). Concentrations of rapamycin (2 μM in DMSO; Sigma-Aldrich), INK128 (0.9 μM in DMSO; Fisher Scientific) and MHY1485 (2μM; Sigma-Aldrich) were determined based on levels in system water that allowed larvae to develop normally from 4dpf-9dpf and whose efficacy was validated through p-S6 immunohistochemistry. Concentrations of dexamethasone (50μM; Sigma-Aldrich) and PLX3397 (1μM; Fisher Scientific) were based on previous studies[19]. The efficacy of Dex has been validated previously in this model [19]. Matched volumes of DMSO were used as a vehicle control for each drug. For multiple day exposures, system water was changed daily and supplemented with fresh compounds.

## BrdU incorporation and TUNEL assays

For bromodeoxyuridine (BrdU) incorporation, larvae were immersed in system water containing 10mM BrdU (Sigma-Aldrich) for 24 hours prior to euthanasia. For terminal deoxynucleotidyl transferase dUTP nick end labeling (TUNEL) assay, the In Situ Cell Death Detection Kit, TMR red (Sigma-Aldrich) assay was used according to manufacturer's instructions to detect apoptotic cells.

## RPE cell isolation and sequencing

Dissected eye tissues were pooled from unablated (MTZ⁻) and ablated (MTZ⁺) DMSO-treated control and rapamycin-treated larvae in triplicate, with each biological replicate consisting of around 80 eyes (40 larvae). Cell dissociation was performed as previously described [19]. Briefly, pooled eyes were dissociated mechanically in a 0.25% trypsin in 1X PBS solution with a 1mL syringe and 27 1/2-gauge needle, strained through a 70uM cell strainer, and spun at 450g for 5 min to pellet the cells. Then, cell pellets were resuspended and washed in 1X PBS three times and resuspended in a final 1X PBS 5% fetal bovine serum solution. Before sorting, propidium iodide (PI) live/dead stain was added and incubated at room temperature for 5 minutes. Sorting gates were established using unstained (GFP⁻ and PI⁻) and single channel controls (GFP⁺/PI⁻; GFP⁻/PI⁺) on a FACSAria IIu cell sorter (BD Biosciences). FACS was performed at the Flow Cytometry Core at the University of Pittsburgh School of Medicine Department of Pediatrics. Sorting gates were maintained for each biological replicate and a maximum of 1000 cells was collected per sample. cDNA was generated using the Smart-seq ultra-Low input RNA kit (Takara Bio USA, Inc.). cDNA quality control was assessed using a TapeStation 2200 system (Agilent Technologies, Inc.). A Nextera XT DNA Library Preparation Kit (Illumina, Inc.) was used to generate indexed 2×75bp paired-end cDNA libraries for subsequent sequencing on NextSeq 500 sequencing platform (Illumina, Inc.). Library preparation, quality control analysis, and next generation sequencing were performed by the Health Sciences Sequencing Core at Children's Hospital of Pittsburgh. 60–100 million reads were obtained per sample.

## Bioinformatics

Bioinformatics analyses were performed on the CLC Genomics Workbench (Qiagen Digital Insights) licensed through the Molecular Biology Information Service of the Health Sciences Library System at the University of Pittsburgh. Raw Illumina reads (FASTQ file format) were imported to CLC Genomics Workbench. Quality checks were performed on imported raw reads and trimmed reads; samples showed average Phred scores of >20. Trimmed paired-end reads were then aligned to the zebrafish reference genome sequence (GRCz11) and differentially expressed genes (DEGs) were determined using filters as follows: 1) log2 fold change absolute value >1; 2) false discovery rate (FDR) p-value <0.05; 3) maximum group mean ≥1. Using the Ensembl gene identifiers of the filtered downregulated DEGs as input parameters, STRING v.11 (https://string-db.org) was used to perform functional pathway enrichment analysis. Reactome datasets were used and the significant enriched Reactome pathways were selected with the thresholds of FDR p-value < 0.05 and gene counts ≥5.

## Quantitative real-time PCR (qRT-PCR)

Two of the three biological cDNA replicates used for RNA-seq were retrieved for qRT-PCR. qRT-PCR reactions were assembled using the iTaq Universal SYBR Green Supermix (Bio-Rad Laboratories). Samples were run in three technical replicates on a CFX384 Touch Real-Time PCR Detection System (Bio-Rad Laboratories). Primers used in this study are included in S10 Table. qRT-PCR data was analyzed using the Livak method [104] to determine delta Ct (ΔCt) values. *beta-actin* was used as the housekeeping gene for expression normalization.

## Immunohistochemistry

Larvae were euthanized with tricaine and fixed in 4% paraformaldehyde (PFA) at room temperature for 2–3 hours or overnight at 4˚C. Fixed larvae were cryoprotected in a 25% then 35% sucrose in 1X PBS and subsequently embedded in optimal cutting temperature (OCT) compound (Uribe and Gross, 2007). Frozen cryosections (12 μm) were cut on a Leica CM1850 cryostat (Leica Biosystems) and mounted onto poly-L-lysine pre-coated glass slides (Superfrost plus; Thermo Fisher). For p-S6 immunostaining, sections were refixed in 4% PFA at room temperature for ten minutes, followed by three washes with 0.1% PBST (1X PBS with 0.1% and Triton X-100). For BrdU immunostaining, after rehydration in 1X PBS, slides were incubated in 4N HCl for 8 minutes at 37˚C for antigen retrieval, and then washed three times with PBTD (1X PBS with 0.1% Tween-20 and 1% DMSO). For general immunostaining, slides were rehydrated in 1X PBS for 5 minutes and then washed three times with PBTD. After blocking with 5% normal goat serum (NGS) in PBTD for at least 2 hours, slides were incubated with either rabbit anti-pS6 (Ser235/236) (1:75; Cell Signaling #4857), rat anti-BrdU (1:250; Abcam; ab6326), mouse anti-ZPR2 (1:250, ZIRC), or mouse anti-mCherry (1:200; Fisher Scientific; NC9580775) antibodies diluted in 5% NGS in PBTD overnight at 4˚C. After three washes with PBTD, slides were incubated with anti-rabbit Alexa Fluor 647 (1:500, Fisher Scientific; A21244), anti-mouse or anti-goat Cy3 (1:250; Jackson ImmunoResearch Laboratories; #115-165-166) secondary antibodies in 5% NGS in PBTD for 2–3 hours at room temperature. After incubation, slides were washed three times with PBTD. DAPI in 5% NGS in PBTD was added onto slides between the first and second washes and slides were incubated in DAPI for 10 minutes. After the third wash, slides were mounted with Vectashield (Vector Laboratories) and sealed with nail polish. All fluorescent images of cryosections were captured using an Olympus Fluoview FV1200 laser scanning microscope and a 40X (1.30 NA) objective (Olympus Corporation).

## Image quantification

Raw z-stack confocal images were processed for quantification using FIJI (ImageJ) [105]. As described previously [19], the number of BrdU+ cells in the RPE layer or CMZ was counted manually on maximum-projected z-stack images using the Cell Counter plugin in FIJI. For quantification of TUNEL+ cells, maximum-projected z-stacks were converted to 8-bit images. Backgrounds were subtracted (rolling ball radius, 60 pixels) from the TUNEL channel. Images were then converted to binary under the same thresholds (60/255) for watershed separation of merged particles. The number of TUNEL+ puncta between the outer plexiform layer and RPE were counted on the segmented images using Analyze Particles in FIJI with the following parameters: size:pixel^2: 5-infinity; circularity: 0.00–1.00. For percent area quantification of p-S6 and mCherry signals in the RPE layer, regions of interest (ROIs) were drawn to encompass the RPE layer using the polygon selection tool in FIJI. ROIs were determined using the DAPI channel to outline the outer edges of photoreceptor outer segments and the bright field channel to outline the basal RPE pigment edges. Thresholding on 8-bit images was performed using either the p-S6 or mCherry channels and the same thresholding parameters were applied to all images in a dataset (40/255 and 50/255, respectively). Pigmentation, eGFP, and ZPR2 were quantified for RPE recovery based on angle measurement by using the angle tool in FIJI. The edges of RPE recovery were determined based on contiguous pigmentation or contiguous expression of eGFP or ZPR2 signals. For retina size measurements, ROIs were drawn using the freehand selection tool in FIJI to encompass the overall neural retina from the inner side of the retinal ganglion cell layer to the outer side of the photoreceptor layer, and area (pixels) was measured from the ROI. The measurements of signal percent area and percentage of RPE recovery were made on the central-most section of each larva. The numbers of BrdU+ cells, TUNEL+ puncta and retina size were the averages from three consecutive central sections per larva.

## Statistics

Statistical analyses were performed using Prism 8 (GraphPad Software). For each dataset, a D'Agostino-Pearson omnibus normality test was performed to determine whether data obeyed normal (Gaussian) distributions. For the datasets that obeyed normal distribution, an unpaired student's t-test with Welch's correction was used for comparisons between two groups, otherwise, nonparametric Mann-Whitney tests were used to determine statistical significance. For analysis where multiple comparisons between three groups were made, Kruskal–Wallis one-way ANOVA with Dunn's multiple comparison test was used to determine variance and significance. For all box plots, top and bottom whiskers represent the maximum and minimum value of each dataset. Median and mean of the dataset were displayed by the line and plus within the box, respectively. Inside triangles represent biological replicates or individual larvae. P-values of >0.05 were considered not significant, and significant p- values were represented as follows: $* \leq 0.05$, $** \leq 0.01$, $*** \leq 0.001$, and $**** \leq 0.0001$. Numbers of independent experiments (N), statistical tests, biological replicates (n), and p-values for each dataset/comparison can be found in S9 Table.

## Supporting information

**S1 Fig. PTU treatment results in low level mTOR activation at 5dpf. (A-B)** Representative fluorescent images of p-S6 staining on cryosections from (A) non-PTU (n = 10) and (B) PTU-treated larvae (n = 10) at 5dpf. (A'-B') Single channel immunofluorescent images of p-S6. Dorsal is up and distal is left. Scale bar = 50μm.
(TIF)

**S2 Fig. MTZ treatment does not elicit mTOR activity in the RPE layer of wild-type larvae.**
(A-D) Immunofluorescent images of p-S6 staining on transverse cryosections from (A,C)
MTZ-treated larvae carrying *rpe65a*:nfsB-eGFP transgene (MTZ+ Tg) and (B,D) MTZ-treated
wild-type larvae (MTZ+ WT). Nuclei (white), p-S6 (magenta), eGFP (green). (A'-D') Single
channel immunofluorescent images of p-S6. (E) p-S6 signals in the RPE layer were signifi-
cantly elevated post-MTZ treatment (6hpi and 2dpi) in *rpe65a*:nfsB-eGFP Tg larvae when
compared to WT controls. Statistical information can be found in S9 Table. Dorsal is up and
distal is left. Scale bar = 50μm.
(TIF)

**S3 Fig. Rapamycin, INK128 and MHY1485 do not affect MTZ-induced apoptosis.** (A-E)
Fluorescent images of TUNEL staining on cryosections from MTZ$^+$ 0.04%DMSO-, 2μM rapa-
mycin-, 2μM MHY1485- and 0.018% DMSO and 0.9μM INK-treated larvae at 1dpi. Single
channel immunofluorescent images of eGFP (A'-E') and TUNEL (A"-E"). (F-J) Fluorescent
images of TUNEL staining on cryosections from MTZ$^-$ 0.04%DMSO-, 2μM rapamycin-, 2μM
MHY1485- and 0.018% DMSO and 0.9μM INK-treated larvae at 6dpf. Quantification of
TUNEL$^+$ puncta between the outer plexiform layer and RPE layer (K) and solely RPE layer (L)
showed no significant differences between DMSO-treated and drug-treated groups. Statistical
information can be found in S9 Table. Nuclei (white), eGFP (green), TUNEL (magenta). Dor-
sal is up and distal is left. Scale bar = 50μm.
(TIF)

**S4 Fig. Pharmacological inhibition of mTOR activity does not affect RPE pigmentation in
unablated larvae.** (A-D) Brightfield representative images of cryosections from MTZ$^-$ DMSO-,
rapamycin- and INK128-treated larvae at 4dpi. Dorsal is up and distal is left. Scale bar = 50μm.
(TIF)

**S5 Fig. Long-term pharmacological inhibition of mTOR signaling impairs retinal growth.**
(A) Quantification of retinal size in larvae treated with DMSO, rapamycin, or INK128 from
4dpf-4dpi (long-term) showed significant decreases in inhibitor-treated larvae from unabated
and ablated groups when compared to the corresponding DMSO-treated controls. (B-E) Fluo-
rescent images on cryosections from DMSO- and rapamycin/INK128-treated larvae at 5dpf.
Yellow lines outline the area of the neural retina for eye size measurement. (F) Quantification
of retinal size of larvae treated with DMSO, rapamycin, or INK128 from 4dpf-5dpf (short-
term) showed comparable overall size between inhibitor- and DMSO-treated groups. p-values:
$^* \leq 0.05$, $^{**} \leq 0.01$. Statistical information can be found in S9 Table. Nuclei (white), eGFP
(green). Dorsal is up and distal is left. Scale bar = 50μm.
(TIF)

**S6 Fig. Pharmacological inhibition of mTOR increased proliferation in the ciliary mar-
ginal zone, regardless of ablation.** Quantification of BrdU$^+$ cells in the ciliary marginal zone
(CMZ) from larvae treated with DMSO, rapamycin, or INK128 from 4dpf - 4dpi showed sig-
nificantly increased cell proliferation in mTOR inhibitor-treated larvae from unabated and
ablated groups, compared with corresponding DMSO-treated controls. p-values: $^* \leq 0.05$, $^{**}$
$\leq 0.01$, and $^{***} \leq 0.001$. Statistical information can be found in S9 Table.
(TIF)

**S7 Fig. RPE, leukocyte, and retinal cells marker gene expression profiles from FACS-sorted
RPE RNA-sequencing datasets.** Heatmap showing average (n = 3) expression values of repre-
sentative RPE, leukocyte, and retinal cell marker genes (including rod and cone photorecep-
tors (PR), bipolar cells (BC), retinal ganglion cells (RGC), amacrine (AC) cells and Muller glia

(MG)) across the different treatment conditions examined by RNA-seq. The RPE markers (*tyrp1a*, *lrata*, *rpe65a*, *tyrp1b*, *dct*, *pmela*, *tyr*, *pmelb*, *best1*) show high expression across all 8 treatment groups relative to other cell type gene markers. The expression values of leukocyte and retinal cell marker genes appear relatively low, with the exception of some photoreceptor genes (*rho*, *crx*, *arr3a*, *arr3b*, *opn1sw1*, *opn1sw2*, *opn1mw1*, *opn1mw2*). Black lines separate marker gene groups for different cell types. Heatmap legend represents log2(counts per million mapped reads (CPM)+1).
(TIF)

**S8 Fig. Quantitative real-time PCR analysis of *scpp8*, *cxcl18a.1*, *lepb*, and *ccn1l1* gene expression to validate 2dpi/7dpf RNA-sequencing results.** qRT-PCR validation of four differentially expressed genes *(scpp8*, *cxcl18a.1*, *lepb*, *ccn1l1)* in MTZ- DMSO, MTZ- Rapamycin, MTZ+ DMSO, and MTZ+ Rapamycin treatment groups used for RNA-seq at 2dpi/7dpf. Similar to RNA-seq results (see Fig 5D and S1 and S6 Tables), data show increased expression of all genes in MTZ+ DMSO groups (green) compared to MTZ- DMSO (tan) and MTZ+ Rapamycin groups (blue). Transcript levels were normalized to the expression of *beta-actin*. Lines represent the median. Each data point represents a biological replicate (n = 2).
(TIF)

**S1 Table. MTZ⁺ 2dpi rapamycin vs. DMSO downregulated genes (top 100).**
(PDF)

**S2 Table. MTZ⁺ 2dpi rapamycin vs. DMSO upregulated genes (top 100).**
(PDF)

**S3 Table. MTZ⁺ 4dpi rapamycin vs. DMSO downregulated genes (top 100).**
(PDF)

**S4 Table. MTZ⁺4dpi rapamycin vs. DMSO upregulated genes (top 100).**
(PDF)

**S5 Table. MTZ⁺ 2dpi rapamycin vs. DMSO enriched downregulated Reactome pathways**
(PDF)

**S6 Table. MTZ⁺ 2dpi DMSO vs. 7dpf MTZ⁻ DMSO upregulated genes (top 100).**
(PDF)

**S7 Table. MTZ⁺ 4dpi rapamycin vs. DMSO enriched downregulated Reactome pathways (top 10).**
(PDF)

**S8 Table. Differential expression of *csf1ra*, *csf1rb* in MTZ⁺ DMSO vs. MTZ⁻ DMSO RPE cells.**
(PDF)

**S9 Table. Statistics.**
(PDF)

**S10 Table. Primers used in this study.**
(PDF)

## Acknowledgments

We are grateful to Kim Gordon for editorial assistance, Joshua Michel for help with FACS, Dr. William MacDonald for help with sequencing, and Dr. Hugh Hammer and his team for zebrafish husbandry.

## Author Contributions

**Conceptualization:** Jeffrey M. Gross.

**Formal analysis:** Fangfang Lu, Lyndsay L. Leach, Jeffrey M. Gross.

**Funding acquisition:** Lyndsay L. Leach, Jeffrey M. Gross.

**Investigation:** Fangfang Lu.

**Project administration:** Lyndsay L. Leach.

**Supervision:** Lyndsay L. Leach, Jeffrey M. Gross.

**Validation:** Fangfang Lu.

**Writing – original draft:** Fangfang Lu.

**Writing – review & editing:** Fangfang Lu, Lyndsay L. Leach, Jeffrey M. Gross.

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
