## [Decision Letter · Decision Letter 0]

29 Jul 2021

Dear Dr Gross,

Thank you very much for submitting your Research Article entitled 'mTOR activity is essential for retinal pigment epithelium regeneration in zebrafish' to PLOS Genetics.

The manuscript was fully evaluated at the editorial level and by independent peer reviewers. The reviewers appreciated the attention to an important problem, but raised some substantial concerns about the current manuscript. Based on the reviews, we will not be able to accept this version of the manuscript, but we would be willing to review a much-revised version. We cannot, of course, promise publication at that time.

If you decide to revise the manuscript for further consideration at PLOS Genetics, please aim to resubmit within the next 60 days, unless it will take extra time to address the concerns of the reviewers, in which case we would appreciate an expected resubmission date by email to plosgenetics@plos.org.

[LINK]

We are sorry that we cannot be more positive about your manuscript at this stage. Please do not hesitate to contact us if you have any concerns or questions.

Yours sincerely,

David R. Hyde, Ph.D.

Guest Editor

PLOS Genetics

Gregory P. Copenhaver

Editor-in-Chief

PLOS Genetics

This manuscript was reviewed by two experts in the field. Both reviewers found several reasons for the publication of this manuscript, including the importance of the zebrafish model, the novelty of ablating RPE cells using the NTR approach, and the identification of a role for mTOR in RPE regeneration. However, both reviewers also identified a number of major concerns with the data presented and conclusions generated from the data. While there is support for publishing this manuscript in PLOS Genetics, it will require a major revision to deal with the numerous major concerns raised by both reviewers. If the authors are interested in addressing these concerns through additional experiments, the journal would welcome the opportunity to rereview the manuscript.

Reviewer's Responses to Questions

**Comments to the Authors:**

Reviewer #1: This paper uses a unique and elegant zebrafish model, where RPE cells are killed by using an NTR approach, to identify a role for mTOR in RPE regeneration. The knowledge they gain understanding the endogenous RPE regenerative response could be helpful in the long-term for the treatment of AMD. Here they extend previous in vitro findings showing that mTOR is required for the survival, migration and proliferation of RPE cells after damage, by moving to an in vivo model. mTOR activity is known to be associated with regeneration in several organs and tissues. The authors show that mTOR is activity in injured RPE, that mTOR inhibition or loss blocks RPE regeneration, and mTOR GOF stimulates the regenerative response. The paper was well written and the data generally nicely presented. The use of both pharmacological inhibitors and an mTOR mutant, the specificity of the NTR model for RPE ablation and its combination with signaling pathway manipulation (LOF and GOF), and the use of transgenic cell specific fluorescent lines are all strengths. These types of approaches are necessary to understand the signaling pathways that control RPE regeneration. Novelty is really restricted to mTOR and the regenerating RPE in vivo, because mTOR is a well-known regulator of regeneration is several organs/tissues, and in vitro data have already implicated mTOR in damaged RPE. While the connection between microglia/macrophages and the mTOR pathway in the RPE after injury is interesting, the data isn’t really there to be able to make strong conclusions about the underlying biology.

Major Points:

1) The authors recently published (Leach et al., 2021) that microglia/macrophages respond to RPE injury, and that they are required for the RPE regenerative response. Here they bring in the mTOR pathway, but the advancement is somewhat limited. They aren’t able to show that it is mTOR signaling in RPE cells that is required for immune cell recruitment, nor that immune cells regulate mTOR activity in RPE cells. Thus, the conceptual advance in terms of immune cell involvement is somewhat limited.

Specifically, there is some question as to the relationship between mTOR signaling, the RPE, and microglia/macrophage, which reduces the strength of their conclusions.

A) The impairment of recruitment of macrophages/microglia to the injured RPE; is that because of a defect in mTOR signaling in the RPE, or could it be that inhibited mTOR signaling in the immune cells themselves affects their recruitment. The data presented here can’t distinguish between the two.

B) The analysis in Figure 7 is confusing, and doesn’t really argue for macrophage/microglial regulation of mTOR signaling in RPE cells. In comparing panel B and C, there is no apparent eGFP signal in panel C; does that not mean there are no RPE cells? So looking at p-S6 label in C and comparing it to B seems inappropriate. If you have fewer RPE (eGFP+) cells, then one would expect less p-S6. These data seem to argue that in the absence of microglia/macrophages there is greater die-back of the RPE (or delayed regeneration?), but I don’t think it says much about p-S6 expression by the RPE, and its regulation by microglia/macrophages. Note that the pink lines are quite dim and hard to see.

2) Not clear to me why in the RNAseq analysis, where RPE cells were FACS-sorted, that there are GABA and neurotransmitter genes that are downregulated. Does this suggest significant contamination of samples with non-RPE neural retina? There should be some assessment of their RNAseq data to indicate how pure it is for RPE cells.

3) The rationale for mTOR inhibitor pre-treatment was not provided, and the pre-treatment seems as though it could complicate the authors ability to say that the effects are due exclusively to manipulating mTOR activity after injury. Because of mTOR inhibitor pre-treatment, I would like to see that the extent of the initial injury in the DMSO and mTOR inhibitor treated groups was comparable. The TUNEL data in S3 doesn’t focus on RPE TUNEL+ cells, in that the methods suggest these data reflect both cells of the outer nuclear layer and the RPE. Thus, it is hard to know if the RPE injuries are comparable, and at 1 dpi the RPE injury is not yet morphologically evident. If the initial injury is larger in the mTOR inhibitor treated group, looking at the regeneration response with two different injuries is not appropriate. Possibly the RPE regenerative response occurs, but is simply delayed with mTOR inhibition, and this could reflect an initial injury difference.

4) Related to the pre-treatment issue raised in point #3, why do the mTOR inhibitor treated eyes (Figures 2C,E,I,K) appear to be much smaller than the DMSO treated counterparts? This does not appear to be the case with the mTOR-/- fish. Does mTOR inhibitor exposure from 4-5 dpf affect eye growth, have non-specific effects, and what consequence could non-RPE related effects of mTOR inhibition have?

Minor comments:

1) In Figure 1 it is unclear why at 3 hpi in the uninjured (MTZ-) larvae that there is considerable p-S6 label. I thought that the injury to the RPE was via MTZ treatment, and so if there is no MTZ there should be no RPE injury and presumably little or no mTOR response? Essentially, why is the 3 hpi and 6-12 hpi labeling different in panels Fig. 1A vs. Fig. 1B-C?

2) Figure 2A experimental design – Rapamycin misspelt

3) The CMZ is highly stimulated after MTZ+ and rapamycin treatment and only minimally so after DMSO treatment. What consequence might this have.

4) BrdU cell counts: BrdU data from uninjured eyes should be included for comparison purposes, especially given the pre-treatment with mTOR inhibitors. Representative images should be provided for the BrdU labeling of injured RPE (Figure 2).

5) Line 182: TUNEL data is in Figure S3 not S2, and the MHY1485 data is Figure S2 not S3.

6) In Figure 3 the embryos is it that PTU-treatment has been stopped, and so pigmentation has returned. Is there no worry that the pigmentation of the RPE could impair visualization (quench fluorescence) of more lightly labeled BrdU+ RPE cells?

7) Figure 3I starts at the origin at 60% recovery. The visually big difference in % recovery is a difference of about 8%. The graph should be redrawn with the origin being at 0%.

8) The BrdU data for the mTOR GOF is separated into a supplemental figure and Figure 4, with only the 3dpi data in Figure 4 being statistically significant. If the argument is that mTOR GOF pushes the peak of RPE proliferation a day earlier, these data should be dealt with together in a single figure. Additionally, this type of time course analysis was not included for the mTOR LOF.

9) The MHY1485 seems to have disruptive effects on the integrity of the INL of treated retinas (e.g Figure S2G, Figure 4I). Is this a concern?

10) The RPE recovery data for MHY1485 was performed at 3 dpi and not 4 dpi as in Figures 2 and 3. Is that because by 4 dpi untreated and treated RPE shows comparable recovery?

11) The statistics in graphs where there are multiple treatment groups, and comparisons made between conditions, these should not all be done as t-tests (Figure 7F, S3F). For instance, in Figure S3F these should not be t-tests, but should be an ANOVA, at least when a single control is used for two treatment groups.

12) How was the mCherry signal measured in Figure 6

13) Line 271: there is no Figure 6G

14) Without the data to support, actual molecular examples of cytokines should be removed from the model in Figure 8.

15) Dexamethasone information needs to be including in Materials and Methods. What is the positive control to show the DEX treatment worked?

16) The BrdU analyses were with a 24 hour pulse, and so it is possible that non-proliferating cells would be counted in the assay.

17) Were measurements done in a manner where the person was blinded to treatment? This type of approach removes subjective bias from measurements, especially when effects are small (e.g. Figure 3I).

18) For the mCherry and p-S6 area measurements more detail needs to be provided in the Material and Methods. How was the ROI chosen? What portion of the eye did it include (RPE only?)?

19) Because these experiments were done on larvae, the authors need to acknowledge that adult RPE could behave somewhat differently.

20) On what basis were the concentrations chosen for drugs, and have they been used in zebrafish previously?

Reviewer #2: This is an interesting manuscript which has potential to reveal novel insights into the mechanism of RPE regeneration in non-mammalian species.

My comments and concerns.

ABSTRACT

-the statement that mTOR activity is sufficient for RPE regeneration needs editing, the evidence may support ability to enhance the process but does not show it is sufficient.

INTRODUCTION

-to give a balanced view on the status of RPE cell based treatments, it is appropriate to mention the effectiveness reported in clinical trials with RPE patches.

RESULTS

-Is p-S6 a specific and selective readout of mTORC1 activation in zebrafish? Is this marker upregulated by other pathways and how do the authors rule these out as driving the p-S6 expression in their model? What is the evidence that the antibody used specifically binds/stains only p-S6 in zebrafish eyes? Has the p-S6 readout been validated by another marker of mTORC1 activation?

-What age and at what time of day does the MTZ induced ablation start? Is this always consistent?

-If MTZ- larvae are unablated, it is confusing in parts that the analysis is referred to as hpi (if not injured)

-There are high levels of pS6+ in the uninjured eyes/RPE that steadily decreases out to 4 days pi (Fig 1 O). Why do such changes occur if there is no intervention/injury? Is p-S6 dynamically expressed in the eye during the eye or during development from 5-9 dpf?

-The levels of marker staining in the RPE is a key measure used to interpret the data. More specific information is needed to know how staining in the RPE is segregated and quantified from other tissues. Furthermore, if there is pS6 staining in macrophages/microglia that have been recruited to the RPE, how is the staining levels attributed to RPE cells?

-In many of the drug treated samples, the eye morphology appears to be regressed compared to controls (e.g. Fig 2E, N, P, 4C, P). This suggests there developmental delays or toxicity induced by the drugs which would confound interpretation of the data. What evidence counters this explanation?

-I'm concerned that the increased levels of marker staining with the mTOR activator (e.g. pS6) at later time points (e.g. 3 dpi) actually reflects a delay/toxic effect to the fish which are now stalled in the 12 hpi stage when the peak occurs in the model. In other words the elevation at 3 dpi is actually due to the fish been delayed and aged around the peak of staining that occurs 12 hpi.

-As RNAseq analysis in zebrafish is notorious for false positives, the authors need to provide data validating the RNAseq by qPCR.

-What QC was performed to confirm that the FACS cells are RPE purified/enriched? Are all the genes known to be expressed in the RPE? Are there any genes identified that are known not to be expressed in the RPE? Are the invading microglia/macrophages excluded from the purified cells?

-In Fig 2, why is there a need for Panel B and D (they appear as the same expt). Same for panel H and J.

-The figures need more details as to the number of biological replicates (fish) and number of independent experiments used.

-In order to make conclusions re inflammation-independent regeneration, the authors needed to demonstrate that Dex gets into the zebrafish eye/ has a biological effect in the eye at this timeline.

**Have all data underlying the figures and results presented in the manuscript been provided?**

Reviewer #1: Yes

Reviewer #2: Yes

PLOS authors have the option to publish the peer review history of their article (what does this mean?). If published, this will include your full peer review and any attached files.

Reviewer #1: No

Reviewer #2: No

---

## [Decision Letter · Decision Letter 1]

27 Nov 2021

Dear Dr Gross,

Thank you very much for submitting your Research Article entitled 'mTOR activity is essential for retinal pigment epithelium regeneration in zebrafish' to PLOS Genetics.

The manuscript was fully evaluated at the editorial level and by independent peer reviewers. The reviewers appreciated the attention to an important problem, but raised some substantial concerns about the current manuscript. Based on the reviews, we will not be able to accept this version of the manuscript, but we would be willing to review a much-revised version. We cannot, of course, promise publication at that time.

While the revised manuscript has been improved and the authors have addressed several of the previous comments, Reviewer 2 makes several important and compelling points that need to be addressed by the reviewers in this manuscript. Please carefully consider the comments of Reviewer 2 and address each experimentally or with a thoughtful rebuttal.

If you decide to revise the manuscript for further consideration at PLOS Genetics, please aim to resubmit within the next 60 days, unless it will take extra time to address the concerns of the reviewers, in which case we would appreciate an expected resubmission date by email to plosgenetics@plos.org.

[LINK]

We are sorry that we cannot be more positive about your manuscript at this stage. Please do not hesitate to contact us if you have any concerns or questions.

Yours sincerely,

David R. Hyde, Ph.D.

Guest Editor

PLOS Genetics

Gregory P. Copenhaver

Editor-in-Chief

PLOS Genetics

Reviewer's Responses to Questions

**Comments to the Authors:**

Reviewer #1: My comments to authors are restricted to the following. The authors have reasonably addressed my concerns.

Reviewer #2: Reviewer #2: This is an interesting manuscript which has potential to reveal novel insights into the mechanism of RPE regeneration in non-mammalian species.

My updated comments and concerns.

RESULTS

-Is p-S6 a specific and selective readout of mTORC1 activation in zebrafish? Is this marker upregulated by other pathways and how do the authors rule these out as driving the p-S6 expression in their model? What is the evidence that the antibody used specifically binds/stains only p-S6 in zebrafish eyes? Has the p-S6 readout been validated by another marker of mTORC1 activation?

p-S6 is a well-established downstream effector of mTORC1 signaling and, yes, it has been used as a readout for mTORC1 activity in zebrafish [20–22], as well as in mice [23,24], chick [25], human tissue [26,27], and in cultured human retinal ganglion cells [28]. S6 is phosphorylated by p70 ribosomal S6 kinase (S6K1 and S6K2) at all serine residues (Ser235, Ser236, Ser240, Ser244, and Ser247) that are subjected to regulation by the mTORC1 signaling pathway. However, several studies described additional protein kinases targeting S6 Ser235 and Ser236 residue independent of of mTORC1/S6K signaling, including p90 ribosomal S6 kinase (RSK), via the extracellular signal-regulated kinase (ERK) signaling [29], protein kinase (PKA) [30], protein kinase B (PKB/AKT) [31], and protein kinase C (PKC)[32], as well as casein kinase 1 targeting S6 residue Ser247 [33]. The p-S6 antibody we used in our study (phospho-S6 Ribosomal Protein (Ser235/236) Rabbit mAb, Cell Signaling #4857) specifically targets ribosomal protein S6 only when phosphorylated at serines 235 and 236, thus we can likely exclude these other pathways that drive p-S6 at other sites.

Rapamycin is a specific inhibitor of the mTOR pathway. It specifically blocks phosphorylation and activation by the p70 S6 protein kinase [34]. In validating this antibody in our study, we demonstrate that p-S6 in the 2dpi RPE was completely abolished by rapamycin, as well as by INK128 treatment (Fig 2A-E), supporting the notion that the p-S6 activity in Fig 1 results from activated mTOR signaling. We also performed negative staining controls, which we stained with p-S6 primary antibody and no secondary antibody, or secondary antibody only, and no signal was detected in the RPE.

***I don’t comprehend how you can “likely exclude” these other pathways if they can also phosphorylate the same Ser residues. Has the p-S6 readout been validated by another marker of mTORC1 activation?

-What age and at what time of day does the MTZ induced ablation start? Is this always consistent?

All ablations in this study were performed at 5dpf, for exactly 24 hours (i.e. larvae are always collected at the precise time from injury indicated). This is described in the Material and Methods and Results/Figures. Ablations are not always performed at the same time of day on different days of experimentation (i.e. on an exact circadian cycle). However, these are generally initiated within a window of a few hours of each other across different experimentation days based on when the animals breed, and always for 24 hours.

***In the methods, state the time window used.

-If MTZ- larvae are unablated, it is confusing in parts that the analysis is referred to as hpi (if not injured)

We apologize for this confusion; these time points are matched to those in the ablated larvae, but we are unsure what the clearest way to label these is. For example, 3hpi is 5.125 dpf, 6hpi is 5.25 dpf, and so on. This seemed cumbersome to use, so we tried to use MTZ- and MTZ+ as the descriptors for uninjured and injured. In this revision, we’ve tried a new way, where we label them as 3hpi UI, with UI defined in the text as uninjured. We don’t know if this makes it clearer or more confusing and are open to any suggestions from a reviewer (reader) as to how best to display these times.

***I agree this is difficult, and that the new labelling is still confusing. A schematic in Fig 1 would clarify the design and the label meaning.

-There are high levels of pS6+ in the uninjured eyes/RPE that steadily decreases out to 4 days pi (Fig 1 O). Why do such changes occur if there is no intervention/injury? Is p-S6 dynamically expressed in the eye during the eye or during development from 5-9 dpf?

Please see our response to R1, minor comment 1, as it also relates to p-S6 levels at 3hpi in the uninjured (UI) eye and PTU treatment/removal. Importantly, the relevant part of our analyses here is that p-S6 levels increase significantly in the MTZ+ eye, relative to those in the uninjured eye, from 6hpi to 3dpi. This, and subsequent studies, show the importance of mTOR signaling during RPE regeneration. Whether mTOR is active dynamically during development, or larval growth, is an interesting question but not one that we pursue in this manuscript.

***Can the authors show that in WT fish, MTZ does not induce higher pS6+?

The levels do increase if there is a direct time-by-time comparison. But can the authors rule out a developmental delay in the MTZ+ fish that would shift the “curve” of expression?

In many of the drug treated samples, the eye morphology appears to be regressed compared to controls (e.g. Fig 2E, N, P, 4C, P). This suggests there developmental delays or toxicity induced by the drugs which would confound interpretation of the data. What evidence counters this explanation?

We addressed this with Reviewer 1, see major comment 4 above. Briefly, we measured retinal size (yellow line in S3 Fig) of larvae treated with DMSO or mTOR inhibitors from 4-5dpf and from 4dpf - 4dpi. These new data are included in S4 Fig. Quantification revealed no significant decrease in retinal size from short-term (4-5dpf) treatment of larvae, but a significant decrease in overall retinal size in larvae from inhibitor-treated groups when compared to the corresponding DMSO-treated controls. This effect occurred regardless of MTZ treatment. These data indicate that short-term mTOR inhibition doesn’t affect eye growth while long-term mTOR inhibition affects eye growth, and that this effect is unrelated to MTZ-induced RPE injury. mTOR plays a number of well known roles in modulating cell size and growth of various tissues and organs [10], including in the Xenopus CMZ [11]. Decreased eye growth after long-term mTOR inhibition might be related with altered cell behavior of retinal stem cells (RSCs) and retinal progenitor cells (RPCs) in the CMZ. mTOR is an essential regulator of stem and progenitor cell behavior but mTOR function within these populations can be different. For example, in spermatogonial stem cells, it was found that TORC1 activation promotes stem cell differentiation at the expense of self renewal [12]. In the Xenopus retina, mTOR inhibition prevents CMZ-derived progenitor cell differentiation, mimicking nutrient deprivation conditions [11]. In the zebrafish retina/CMZ, it is possible that long-term mTOR inhibition also leads to attenuation of CMZ-derived RPC differentiation to retinal neurons, while increasing RSCs for self-renewal. Indeed, in support of this notion, mTOR inhibition results in increased proliferation within the CMZ (see R1 Minor Comment 3 below and S5 Fig). A similar mechanism has also been recently identified in hematopoietic stem cells where mTOR inhibition results in a loss of quiescence of hematopoietic stem cells and increased proliferation [13]. We have added additional text to this point in the manuscript; however, as our focus here is on RPE regeneration, we do not pursue these CMZ effects further in this study, but it is one of interest for us in future studies focused on CMZ maintenance.

***While I appreciate the comprehensive response, I don’t consider that this definitively locks affects of the drugs to changes in RPE regeneration. The assay endpoint is as 9 dpf, so even if the drugs have no obvious effect on eye development from 4-5 dpf, they do from 4-9 dpf. If there is a “general ocular toxicity” with the drugs that may explain the reduced BrdU levels in the RPE at 9 dpf, and not a specific effect on mTOR in RPE regeneration. Have the authors shown that if they add the mTOR drugs late (e.g. 1-2 dpi) that they still reduce BrdU levels in the “regenerating RPE”?

-As RNAseq analysis in zebrafish is notorious for false positives, the authors need to provide data validating the

RNAseq by qPCR.

We’re aware of no published evidence that RNA-seq in zebrafish is more prone to false positives than in other systems/models. We also respectfully disagree about the need for additional validation of our RNA-seq dataset. This is a hypothesis generating experiment, not an end result, and we carefully tested/validated one hypothesis resulting from our data - that immune responses were modulated by the mTOR pathway. There are many additional hypotheses that come from this dataset, each of which needs to be carefully validated by experiments specific to the gene or pathway involved. Here, we include three biological replicates for each group and we filtered (log2 fold change>1; FDR p-value<0.05, max group mean≥1) DEGs to exclude the genes with small differences in expression. The issue of needing additional validation is one that we have thought of quite a lot over the last few years, as most of our studies involved using RNA-seq to generate hypotheses. It is also one that has been addressed in a number of recent editorials as well [35]. It is not clear to us that qPCR validation provides any additional rigor to a study unless there are insufficient replicates or a model is based on differential expression of only a few genes with low expression levels and/or with differences in expression that are quite small. Put differently, we don’t see how simply picking a few DEGs and doing qPCR is a robust validation system, nor how it addresses the data set as a whole where hundreds to thousands of transcripts change expression under each condition and at varying statistical thresholds. Philosophically (and probably at the risk of opening a larger debate that is unrelated to the contents of this manuscript, which is not our intent), we question whether small changes that are validated as statistically significant in a dataset are in fact biologically relevant and important, which can really only be determined experimentally with a thorough characterization. One can easily throw in a qPCR graph showing that five “candidate” genes validate in the direction and approximate scale predicted by the RNA-seq data, but in a large data set with many candidates, this cannot engender any more confidence as to the fidelity of the data set as a whole nor does it have much meaning biologically without further experimentation (an issue nicely addressed in [36]). There are obvious caveats to that statement, but our point is that by simply cherry picking a few candidates and showing that they behave in qPCR as predicted by the RNA-seq adds no additional rigor or biological insight to a story. We selected one candidate pathway here and rigorously validated that experimentally as important and this is the type of follow-up study that is necessary for any putative candidate gene/pathway to be “validated”.

***I agree that qPCR validation of a small number of genes may have little meaning biologically. However, I am aware from others that there are issues with mis-mapping of reads in zebrafish RNAseq data sets. In order, to provide technical validation of the dataset so that the community can be confident in using it as a point to test hypotheses, I think it is very reasonable to show validation of a small cohort of genes. If they validate, as you expect, that provides confidence to everyone.

-In order to make conclusions re inflammation-independent regeneration, the authors needed to demonstrate that Dex gets into the zebrafish eye/ has a biological effect in the eye at this timeline.

The efficacy of Dex has been validated previously in this injury/regeneration model by showing the upregulated expression of pxr, a transcriptional target of Dex, after one-day exposure to Dex [1]. Details on Dex concentration and manufacturer information can be found in the Materials and Methods section.

***The paper cited does show increased pxr after Dex treatment, but this was in larvae, not in eyes, and therefore does not adequately address the original comment.

**Have all data underlying the figures and results presented in the manuscript been provided?**

Reviewer #1: **No: **As best I can tell the authors state "all data are fully available without restriction" but do not provide details about how to obtain these data (numerical data that underlies graphs). An accession number in GEO is provided for the RNAseq data.

Reviewer #2: Yes

PLOS authors have the option to publish the peer review history of their article (what does this mean?). If published, this will include your full peer review and any attached files.

Reviewer #1: No

Reviewer #2: No

---

## [Decision Letter · Decision Letter 2]

23 Feb 2022

Dear Dr Gross,

We are pleased to inform you that your manuscript entitled "mTOR activity is essential for retinal pigment epithelium regeneration in zebrafish" has been editorially accepted for publication in PLOS Genetics. Congratulations!

Yours sincerely,

David R. Hyde, Ph.D.

Guest Editor

PLOS Genetics

Gregory P. Copenhaver

Editor-in-Chief

PLOS Genetics

Comments from the reviewers (if applicable):

Reviewer's Responses to Questions

**Comments to the Authors:**

Reviewer #2: I commend the authors for the work they have undertaken to address my concerns including the addition of new data.

I appreciate their frustration in relation to qPCR validation. My comments were not directed to them, but concerns about zebrafish RNAseq datasets more generally. Indeed, what the authors have done here is to provide addition evidenec to rebut any such concerns.

**Have all data underlying the figures and results presented in the manuscript been provided?**

Reviewer #2: Yes

PLOS authors have the option to publish the peer review history of their article (what does this mean?). If published, this will include your full peer review and any attached files.

Reviewer #2: No

**Data Deposition**

http://datadryad.org/submit?journalID=pgenetics&manu=PGENETICS-D-21-00709R2

**Press Queries**

---

## [Editor Report · Acceptance letter]

7 Mar 2022

PGENETICS-D-21-00709R2 

mTOR activity is essential for retinal pigment epithelium regeneration in zebrafish 

Dear Dr Gross, 

We are pleased to inform you that your manuscript entitled "mTOR activity is essential for retinal pigment epithelium regeneration in zebrafish" has been formally accepted for publication in PLOS Genetics! Your manuscript is now with our production department and you will be notified of the publication date in due course.

With kind regards,

Anita Estes

PLOS Genetics

On behalf of:
